# Global and regional perspectives on optimizing thermo-responsive dynamic windows for energy-efficient buildings

Yuan Gao ✉, Jacob C. Jonsson, D. Charlie Curcija, Simon Vidanovic & Tianzhen Hong

Architectural thermo-responsive dynamic windows offer an autonomous solution for solar heat regulation, thereby reducing building energy consumption. Previous work has emphasized the significance of thermo-responsive windows in hot climates due to their role in solar heat control and subsequent energy conservation; conversely, our study provides a different perspective. Through a global-scale analysis, we explore over 100 material samples and execute more than 2.8 million simulations across over two thousand global locations. World heatmap results, derived from well-trained artificial neural network models, reveal that thermo-responsive windows are especially useful in climates where buildings demand both heating and cooling energy, whereas thermo-responsive windows with optimal transition temperatures show no dynamic features in most of low-latitude tropical regions. Additionally, this study provides a practical guideline and an open-source mapping tool to optimize the intrinsic properties of thermo-responsive materials and evaluate their energy performance for sustainable buildings at various geographical scales.

Buildings represent about 36% of the global primary energy demand, and about 37% of global energy-related carbon dioxide ($CO_2$) emissions[1,2]. Windows are often identified as one of the least energy-efficient components of a building, contributing to ~30% of the energy loss associated with heating and cooling systems[3]. Consequently, the incorporation of energy-efficient windows presents the potential for both new constructions and retrofit projects to meet energy-saving objectives. This is particularly relevant as energy demand rebounds in the post-pandemic era[4], alongside the escalation of extreme weather events[5] and the implementation of stringent decarbonization policies[2,6]. Thermo-responsive (TR) dynamic windows, representing an efficient solution among passive smart window technologies, provide cost-effective energy management[7]. TR windows possess the ability to autonomously modulate solar heat gain in accordance with the temperature of the thermo-responsive materials or integrated sensors[8]. This self-regulating mechanism, requiring no manual intervention, curtails the energy expended on heating, ventilation, and air conditioning (HVAC) systems[9,10].

TR dynamic windows influence building energy consumption with three critical properties: clear-state solar transmittance ($\tau_{clear}$), dark-state solar transmittance ($\tau_{dark}$), and transition temperature ($T_{tran}$). Such thermo-responsive features can be realized by directly incorporating a layer of thermochromic (TC) material into the window glazing. A wide variety of TC materials have been investigated for their potential application in dynamic windows. Vanadium dioxide ($VO_2$) is one of the most studied TC materials owing to its reversible thermal phase transition, which occurs between an insulating monoclinic phase at lower temperatures and a metallic tetragonal rutile phase at higher temperatures[11]. The intrinsic transition temperature ($T_{tran}$) of $VO_2$ is 68 °C; however, this can be adjusted to near room temperature through methods such as defect control, elemental doping, and nano- and microscale engineering, which concurrently influence its optical and electrical properties[11]. The primary modulation of transmittance mainly occurs in the near-infrared (NIR) range, ensuring a stable and uniform color appearance; however, resulting in a limited regulation of

Building Technology & Urban Systems Division, Lawrence Berkeley National Laboratory, Berkeley, CA, USA. ✉e-mail: y.gao@lbl.gov

solar transmittance ($\Delta\tau_{sol}$, typically below 20%). Thermochromic hydrogels, compared with $VO_2$, exhibit higher $\Delta\tau_{sol}$ (as high as above 80%) due to their reversible hydrophilic/hydrophobic phase transition. This transition is driven by the breaking and forming of hydrogen bonds, which causes the polymer chains to alternate between expanded hydrophobic associations above $T_{tran}$ and hydrogen-bonded interactions with water molecules below $T_{tran}$[12–16]. Poly(N-iso-propylacrylamide) (PNIPAM), one of the most widely studied thermochromic hydrogels, undergoes a sharp phase transition around 32 °C, which is also tunable approximately between 25 and 40 °C[17]. Other TC materials, such as liquid crystal[18,19], ionogels[20], polymers[21], elastomers[22], perovskite[23–26], and hybrid materials[27], are also promising for smart window applications. Though different TC materials show various electrical and optical properties, material scientists seem to reach a consensus that the optimal $T_{tran}$ should be tuned to near room temperature[10], which is an unsubstantiated conclusion.

Alternatively, TR dynamic windows can be achieved by integrating temperature sensors with electrochromic (EC) windows, enabling passive, automatic control in addition to manual management[28]. In such control systems, $T_{tran}$ is programmable and easier to adjust compared with TC windows. Traditional EC materials refer to transition metal oxides, such as tungsten oxide ($WO_3$), molybdenum oxide ($MoO_3$), vanadium oxide ($V_2O_5$), etc., however, suffering from low transition speed, unsatisfied optical modulation, and poor color rendering index (CRI)[29]. Recent study in synthesizing porous bilayer hybrid $WO_3$ nanoarrays improved the solar modulation ($\Delta\tau_{sol} > 0.8$) and response speed (3.0 s for coloring and 3.6 s for bleaching)[30]. Additionally, electronically tintable windows based on reversible metal electrodeposition (RME) attract more attention due to their high $\Delta\tau_{sol}$ (about 0.7) and nearly zero $\tau_{dark}$[31,32]. Others electrochromic technologies, such as suspended particle devices (SPDs), polymer dispersed liquid crystal (PDLC), microshutters, etc.[33], show various optical and electrical properties, requiring customized $T_{tran}$ to automatically control EC windows for energy reduction in buildings.

Though numerous TC and EC materials and devices have been developed, two major challenges remain for their implementation in TR dynamic windows. First, pinpointing an optimal $T_{tran}$ proves challenging due to its sensitivity to various factors, including intrinsic optical properties, climate conditions, building configurations, etc. Previous building simulation studies suggested the optimal $T_{tran}$ falling between 16 and 27 °C, which is, however, concluded by several independent studies utilizing different building models, specific material properties and limited representative cities[9]. The other challenge is the lack of effective indicators to identify suitable scenarios for TR dynamic windows. Previous simulation results frequently demonstrated greater energy savings by such windows in hot climates than in cold climates[9]. However, this does not demonstrate the superiority of TR dynamic windows over static window technologies in hot climates, especially without quantifying the dynamic nature. Therefore, innovative indices are needed to evaluate the suitability of TR dynamic windows in a more comprehensive manner.

In this work, we elucidate the energy-saving potential of TR windows and explore the correlations between climate suitability and intrinsic material properties on a worldwide scale. A comprehensive understanding of such correlations is important for optimizing dynamic window materials that suit diverse weather conditions or customizing materials and dynamic windows for specific locations. By summarizing key properties of over 100 materials, and analyzing data from over 2.8 million simulation runs, we propose an innovative recommendation index for TR windows, factoring in both energy savings and necessity level. World heatmaps of indicators for TR dynamic windows have been obtained by artificial neural networks trained by data from over two thousand global weather conditions. Our analysis reveals that TR windows with optimal transition temperatures lack dynamism in most low-latitude tropical regions,

functioning similarly to static windows in terms of energy savings. Seasonal dynamics of TR windows appear more important for energy saving compared with daily fluctuations. High clear-state solar transmittance is favorable for energy conservation, and it strongly influences the optimal transition temperature, which also depends on climate conditions and other building configurations. As a practical contribution, we offer a comprehensive guideline and an open-source mapping tool for researchers to identify ideal regions for the application of their TR materials and window products.

## Results

### Optimal key parameters for maximum building energy saving

A functional TR dynamic window incorporates TC or EC materials (Fig. 1a), which have been widely explored and studied[9,28,33], yet a fundamental understanding crucial for optimizing these materials remains elusive. Regardless of the employed materials, three key parameters of TR windows, $\tau_{clear}$, $\tau_{dark}$, and $T_{tran}$, are essential for effective solar heat regulation and energy saving (Fig. 1b). To gain a comprehensive understanding of the expansive scope of the key parameters, this study examines over 100 TC and EC samples across six material categories, as documented in recent literature (Supplementary Table 1). Figure 1c shows the solar transmittance domain of various TC and EC materials, indicating the flexibility of utilizing different transmittance combinations for target building applications. Hydrogel exhibits the highest solar transmittance in the clear state with a nearly zero dark-state transmittance, whereas materials like $VO_2$ reveal a notable correlation between $\tau_{dark}$ and $\tau_{clear}$. Additionally, TR windows based on $WO_3$ nanoarrays and RME demonstrate impressive solar modulation capabilities, surpassing those of traditional transition metal oxides. Compared to the readily adjustable $T_{tran}$ for EC materials within feedback control systems, the $T_{tran}$ for TC materials have an intrinsic characteristic that, despite being tunable, remains fixed for a given material. Figure 1d illustrates the broad range of $T_{tran}$ for TC materials, alongside their capacities for solar modulation, suggesting that, while material scientists have the ability to adjust the $T_{tran}$ for various TC materials, there is a lack of clarity regarding the optimal placement of $T_{tran}$ values. The $T_{tran}$ values of $VO_2$ and hydrogel predominantly center around 68 and 32 °C, respectively, and have undergone adjustments to approach room temperature. Meanwhile, hydrogel demonstrates a greater capability for solar modulation compared to $VO_2$. Liquid crystals primarily transition from clear to scattered states instead of darkening, resulting in a visual change that is more noticeable than $\Delta\tau_{sol}$.

To provide a comprehensive understanding and guide the application of TR windows, we utilize computer simulations that cover a wider range of material properties, windows and building configurations, and weather conditions, offering deeper insights compared to field experiments[34]. Such all-encompassing simulations lead to three simulation batches, totaling ~2.8 million cases (Supplementary Table 2, Supplementary Figs. 1–48). After excluding the less critical variables through statistical analysis, we focus on the energy performance of a clear-clear double-pane south-facing window with dynamic material applied on Surface 1 within the prototypical office space without lighting control (LC No) (Supplementary Fig. 49) in three US cities (Phoenix, Baltimore, and Minneapolis), representing typical climates (hot, mixed, and cold)[35,36]. Figure 1e–g shows the total site energy saving per conditioned building (floor) area ($\Delta E_{total}$) as a function of $T_{tran}$ for three typical climates. The $\Delta E_{total}$ in hot climate (Fig. 1e) shows distinct patterns compared with those in the mixed and cold climates (Fig. 1f, g). The maximum $\Delta E_{total}$ plateau at low $T_{tran}$ in the hot climate (no heating demand) indicates that the TR window saves the most building energy when it constantly stays in the dark state (Fig. 1e). With $T_{tran}$ at (or below) this optimal value (11 °C for the blue line which represents $\tau_{clear} = 0.4$, and $\tau_{dark} = 0$), the TR window will perform equally as a static window. Therefore, the maximum $\Delta E_{total}$ is achieved at the minimum $\tau_{dark}$ and a sufficient low $T_{tran}$ regardless of $\tau_{clear}$ in the

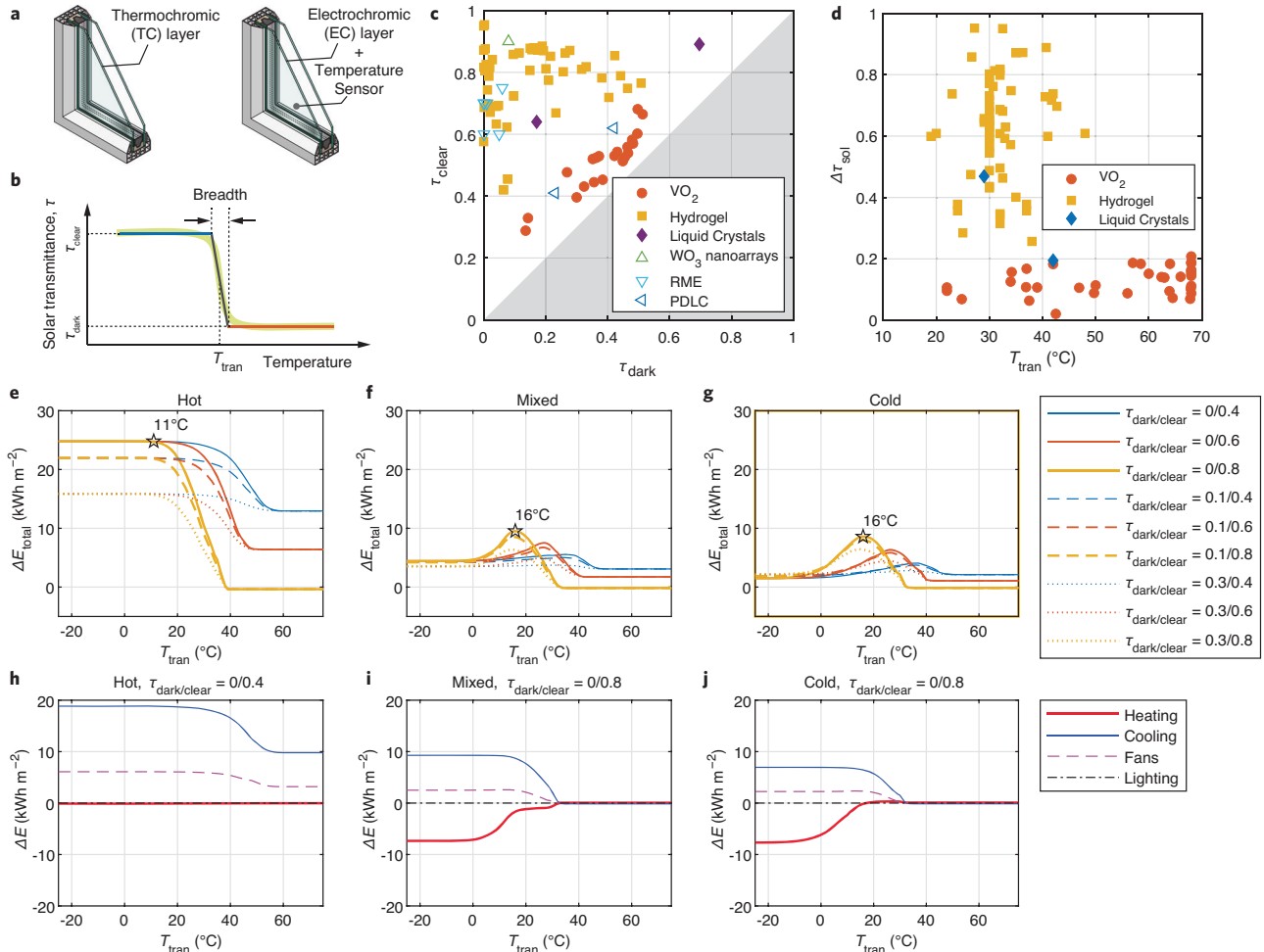

**Fig. 1 | Thermo-responsive (TR) dynamic windows: configurations, key parameters and statistics, and representative energy performance. a** Double-pane TR window configuration. **b** Key parameters of TR windows. **c** Solar transmittance for dynamic window materials in clear ($\tau_{clear}$) and dark ($\tau_{dark}$) states. The gray shading area represents the excluded region, where $\tau_{clear} < \tau_{dark}$. RME indicates Reversible metal electrodeposition, and PDLC indicates polymer dispersed liquid crystal. **d** Solar modulation ($\Delta\tau_{sol}$) and transition temperature ($T_{tran}$) for thermochromic (TC) materials. **e–g**, Total site energy savings per conditioned building (floor) area ($\Delta E_{total}$) by TR windows as the function of transition temperature in hot (**e**), mixed (**f**), and cold (**g**) climates, respectively. **h–j** Energy savings ($\Delta E$) by TR windows (with optimal properties) in the aspect of heating, cooling, fans, and lighting as the function of transition temperature in hot (**h**), mixed (**i**), and cold (**j**) climates, respectively. Source data are provided as a Source Data file.

hot climate. By contrast, obvious $\Delta E_{total}$ peaks are observed at the optimal $T_{tran}$ for the mixed (Fig. 1f) and cold (Fig. 1g) climates, where both heating and cooling are needed. Those peaks result from the sum of heating, cooling, fans, and lighting energy (Fig. 1h, i). A large modulation of solar transmittance favors building energy saving. Specifically, a low $\tau_{dark}$ decreases the cooling energy; and a high $\tau_{clear}$ lowers the heating energy.

Compared to the buildings without lighting control (LC No), we observed higher optimal $\tau_{dark}$ and $T_{tran}$ in buildings with lighting control (LC Yes) because extra solar radiation reduces the artificial lighting energy (Supplementary Figs. 50–65). In addition to the aforementioned office spaces, we further validate our findings through the Department of Energy (DOE) prototype building models of a three-floor medium-sized office and a four-floor medium-rise residential apartment (Supplementary Figs. 66, 67), reinforcing the conclusions derived from the simplified model. Other comparisons with NIR-response windows and passive buildings can also be found in Supplementary Figs. 68–71.

## Sensitivity analysis of optimal transition temperature
An optimal transition temperature is the key for TR windows to dynamically balance the heating and cooling energy in a building. The

optimal $T_{tran}$ can be sensitive to both intrinsic material properties and environmental parameters (window and building configurations, climate types, etc.). Therefore, it is vital to clarify the conditions under which the optimal $T_{tran}$ is obtained. Here, we use the star-marked optimal $T_{tran}$ in Fig. 1 as the base case and investigate the sensitivity of the optimal $T_{tran}$ by varying one parameter or restriction at a time (Fig. 2).

Relaxing the energy saving ($\Delta E_{total}$) requirement will allow the TR windows to have a wider tolerance of $T_{tran}$, i.e., a $T_{tran}$ that is higher than the optimal will be acceptable when we are not pursuing the maximum $\Delta E_{total}$. In the hot climate, the upper limit of $T_{tran}$ can be extended to 44 °C when $\Delta E_{total}$ is 5 kWh m$^{-2}$ less than the maximum (Fig. 2a). Compared to the hot climate, such an upper limit is lower for mixed (26 °C) and cold (27 °C) climates (Fig. 2b, c). TR layer applied to the exterior surface shows a lower $T_{tran}$ than the interior surface regardless of climate types. Changing the optical properties ($\tau_{clear}$ and $\tau_{dark}$) also affects the optimal $T_{tran}$ (Fig. 2d–f). The optimal $T_{tran}$ is more sensitive to $\tau_{clear}$ than $\tau_{dark}$. In the hot climate, lower $\tau_{clear}$ results in higher optimal $T_{tran}$, but will not affect the $\Delta E_{total}$ because the TR window barely turns clear. In mixed and cold climates, the optimal $T_{tran}$ can be up to nearly 40 °C when $\tau_{clear}$ equals to 0.4, which, however, results in a decreased $\Delta E_{total}$ due to the extra heating energy consumption compared to high $\tau_{clear}$ (0.8). Other sensitivity analyses

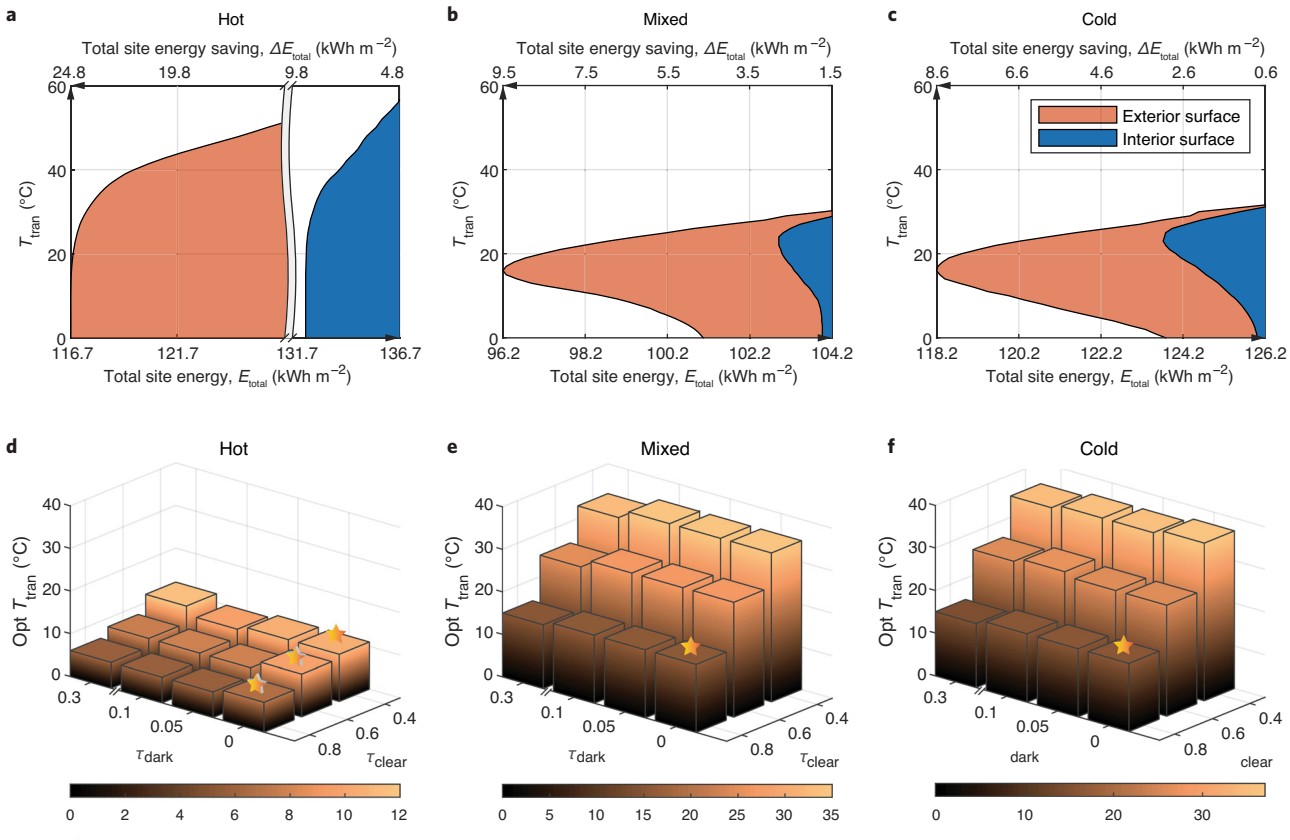

**Fig. 2 | Sensitivity analysis of optimal transition temperature for thermoresponsive (TR) windows. a–c** Tolerance range of transition temperature ($T_{tran}$) as the energy saving requirement is relaxed when TR material is applied on exterior and interior surfaces in hot (**a**), mixed (**b**), and cold (**c**) climates, respectively. **d–f** Three-dimensional bar chart of optimal transition temperature (opt $T_{tran}$) as the function of clear- and dark-state solar transmittance of TR windows in hot (**d**), mixed (**e**), and cold (**f**) climates, respectively. Source data are provided as a Source Data file.

on building configurations, such as window-to-wall ratio, electrical equipment level, heating and cooling setpoints, etc., can be found in Supplementary Fig. 72.

**Proposed indices for TR window evaluation**

Statistic results show that seasonal dynamics of TR windows are more obvious than daily changes between states (Supplementary Fig. 73). Cooling energy is still in need even in cold weather because the office room temperature is kept within a thermally comfortable range, and can easily reach the cooling setpoint under strong solar radiation during summer daytime. To quantify the benefit of the dynamic properties under different climate conditions, here we introduce several definitions, which are illustrated with typical curves of building energy consumption ($E_{total}$) and energy saving ($\Delta E_{total}$) as the function of transition temperature ($T_{tran}$) in Fig. 3a. Here, we define $\Delta E_{TR}$ as the total site energy saving of buildings using TR windows. In the following context (Figs. 3, 4), $\Delta E_{TR}$ specifically indicates the maximum potential of building energy saving by TR windows with optimal $\tau_{clear}$, $\tau_{dark}$ and $T_{tran}$ unless stated otherwise (e.g., Fig. 5 and Supplementary Figs. 81–116). The corresponding building energy consumption is denoted by $E_{TR}$. We calculated $\Delta E_{TR}$ in 16 representative US cities and rendered a heatmap of $\Delta E_{TR}$ that covers the US climate classifications[35]. As shown in Fig. 3b, a darker color can be found in hotter areas, indicating that the TR windows have a greater potential of saving building energy in hot climates than in cold climates.

We also define $E_{dark}$ and $E_{clear}$ as the $E_{total}$ by constantly dark (low $T_{tran}$) and constantly clear (high $T_{tran}$) TR windows, respectively. The smaller between $E_{dark}$ and $E_{clear}$ is denoted by $E_{static}$, which stands for

the $E_{total}$ by a static window that is equivalent to the TR window without the dynamic feature. We then define $\Delta E_n$ as the difference between $E_{static}$ and $E_{TR}$, indicating the necessity level of using TR windows compared to static windows (Fig. 3a). Figure 3c shows that contrary to $\Delta E_{TR}$, the necessity level of using TR windows in most of hot climates is as low as zero, which agrees with the $\Delta E_{total} - T_{tran}$ curves in a typical hot weather (Fig. 1d). A zero $\Delta E_n$ indicates that a best-performing TR window saves the same building energy as the equivalent static window. Note that $\Delta E_n$ is a nonnegative value when TR windows have the optimal $T_{tran}$ due to the nature of the definition. However, $\Delta E_n$ could be a negative value if $T_{tran}$ is not optimal, indicating that a static window might outperform a TR window in certain climates.

Considering both energy saving and necessity level, we define the recommendation index of using TR windows (TRRI) as the product of $\mathrm{ReLU}(\Delta E_{TR})$ and $\mathrm{ReLU}(\Delta E_n)$, where ReLU indicates the rectified linear unit, which is a piecewise linear function that outputs the input directly if it is positive, otherwise, outputs zero. A nonpositive $\Delta E_{TR}$ or $\Delta E_n$ leads to zero TRRI, indicating that TR windows are unnecessary when there is no energy saving or optical dynamics. A higher TRRI indicates a stronger recommendation for using TR windows. As shown in Fig. 3d, the darkest color lies in the cold area, where $\Delta E_n$ is relatively higher, though $\Delta E_{TR}$ is of medium value. In general, cold and mixed climates show higher TRRI than hot climates. Our simulation results show that cooling is necessary for the reference commercial office where room temperature is kept between heating and cooling setpoints, even in cold climates. This fact indicates that the dynamic feature of TR windows benefits the energy saving of buildings in mixed and cold climates, where both heating and cooling are needed. In hot climates,

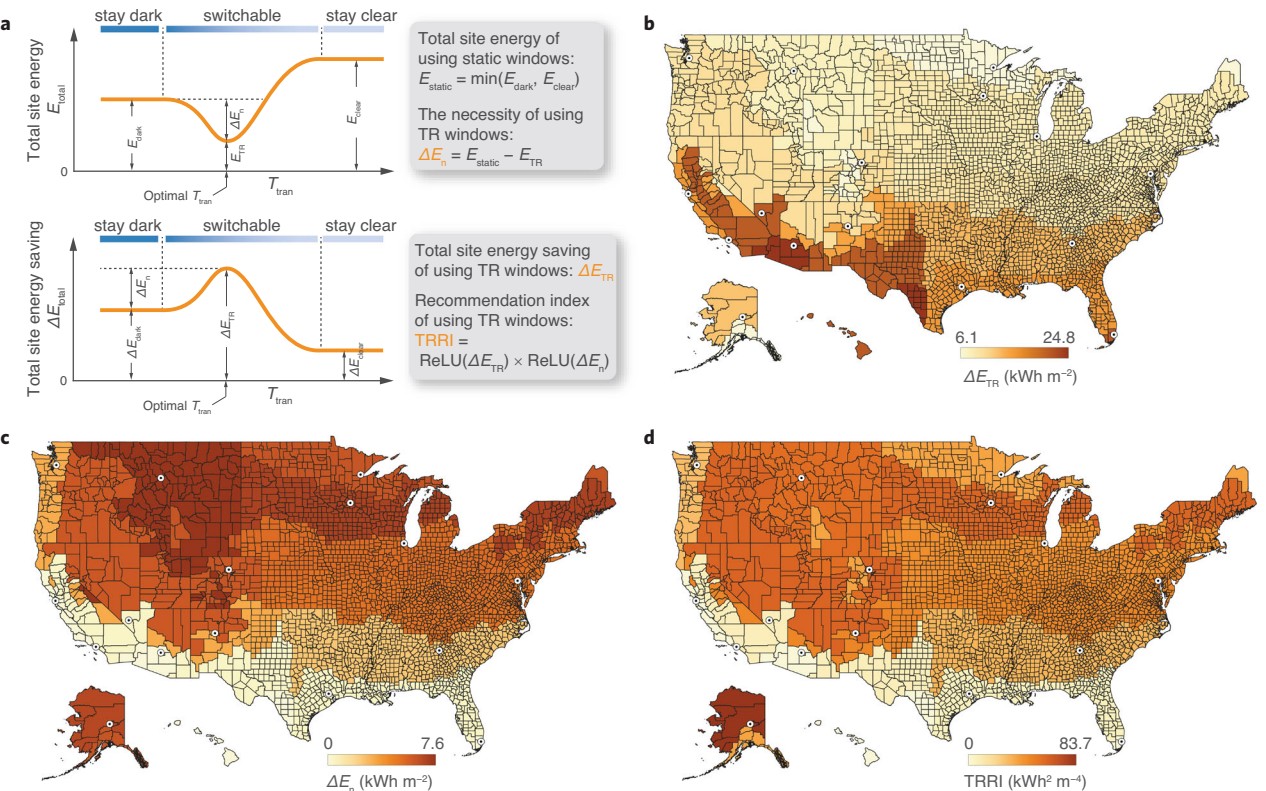

**Fig. 3 | Innovative indicators for thermo-responsive (TR) window performance.** **a** Definition of key indicators about using thermo-responsive (TR) windows, total site energy saving ($\Delta E_{TR}$), necessity level ($\Delta E_n$), and recommendation index (TRRI), to comprehensively evaluate the performance of TR windows. $E_{dark}$ and $E_{clear}$ indicate the total site energy ($E_{total}$) by constantly dark and constantly clear TR windows, respectively. **b–d** US heatmaps of $\Delta E_{TR}$ (**b**), $\Delta E_n$ (**c**), and TRRI (**d**) rendered by representative cities. Maps of the US include Alaska and Hawaii as insets in the bottom left corner. The main map covers the contiguous states (25°N–49°N, 67°W–125°W), with Alaska and Hawaii positioned for visual clarity. State and county boundaries are provided by the US Census Bureau. Source data are provided as a Source Data file.

however, the key feature of TR windows, i.e., the optical transition between clear and dark states, becomes useless regarding building energy saving.

## Global and regional performance evaluation

In the US heatmaps (Fig. 3b–d), each climate classification zone is roughly rendered by the parameter values of the corresponding representative cities. Acquiring high-resolution global heatmaps using traditional methods, namely simulating and calculating the TRRI for each pixel, would demand significant computational resources and time. To respond to this challenge, we propose to train artificial neural networks (ANNs) for TRRI by inputting geographic (latitude) and weather (solar radiation and air temperature) data that can be extracted from weather files and is publicly available as the information in a world-map format[37,38]. Instead of directly training a single ANN for TRRI, we introduce intermediate parameters that have direct physical or mathematical interactions with the inputs (geographic and weather data) or output (TRRI). We reason that the global vertical irradiance (GVI) is more directly related to the window performance compared with other commonly used solar radiation indices, e.g., direct normal irradiation (DNI), global horizontal irradiation (GHI), diffuse horizontal irradiation (DHI), etc. We build the first ANN to obtain GVI from the inputs of latitude (Lat), DNI, GHI, and DHI. Then, we use GVI as well as Lat, the average, minimum, and maximum air temperature ($T_{avg}$, $T_{min}$, and $T_{max}$) as the input of ANNs for $\Delta E_{TR}$ and $\Delta E_n$, respectively. In the end, TRRI can be easily calculated by the product of $ReLU(\Delta E_{TR})$ and $ReLU(\Delta E_n)$ (Supplementary Fig. 74). Weather data from over two thousand stations covering most of the Köppen climate classifications in the world (Supplementary Fig. 75) are

used in the EnergyPlus simulations and ANNs training. The good linear regressions between target and prediction indicate that the trend of TRRI can be well predicted from the inputs of latitude, solar radiation, and air temperature (Fig. 4a–c). Detailed ANN performance and hyperparameter tuning process can be found in Method.

We then obtain the world heatmaps of $\Delta E_{TR}$, $\Delta E_n$ and TRRI (Fig. 4e–g) by inputting world-map-format matrices of Lat, DNI, GHI, DHI, $T_{avg}$, $T_{min}$, and $T_{max}$ to three properly trained and validated ANNs for GVI, $\Delta E_{TR}$, and $\Delta E_n$ (Supplementary Figs. 76–78). The heatmaps and corresponding latitude charts (Supplementary Fig. 79) indicate that low-latitude zones (0° to 23.5°N and S) tend to have zero TRRIs, and positive TRRIs mainly located in the middle-latitude zones (23.5° to 66.5°N and S). $\Delta E_{TR}$ peaks at about 30°N and S, where the optimal $T_{tran}$ reaches the lowest point. Because building distributions are highly related to population density, we refine the latitude charts by weighting the pixel counts on the heatmap according to the corresponding population density values (Supplementary Figs. 80, 81)[39]. Consequently, regions with higher population densities exhibit a denser scattering of red dots ($\Delta E_n > 0$), offering a more nuanced representation of practical $\Delta E_{TR}$ and recommended $T_{tran}$. Besides the optimal situation, we further trained ANNs for $\Delta E_{TR}$, $\Delta E_n$, and optimal $T_{tran}$ under different $\tau_{clear}$ and $\tau_{dark}$ conditions (Supplementary Figs. 82–121). Using the resultant ANNs, we can predict the global distribution of $\Delta E_{TR}$, $\Delta E_n$, and optimal $T_{tran}$ by the given $\tau_{clear}$ and $\tau_{dark}$ of TR materials (Fig. 4d, h).

Ultimately, we trained two ANNs for $\Delta E_{TR}$ and $\Delta E_n$, respectively, by further including $\tau_{clear}$, $\tau_{dark}$, and $T_{tran}$ as the inputs (Supplementary Figs. 122–124). Such ANNs can generate the global heatmaps of $\Delta E_{TR}$, $\Delta E_n$, and TRRI by inputting the three key properties of TR

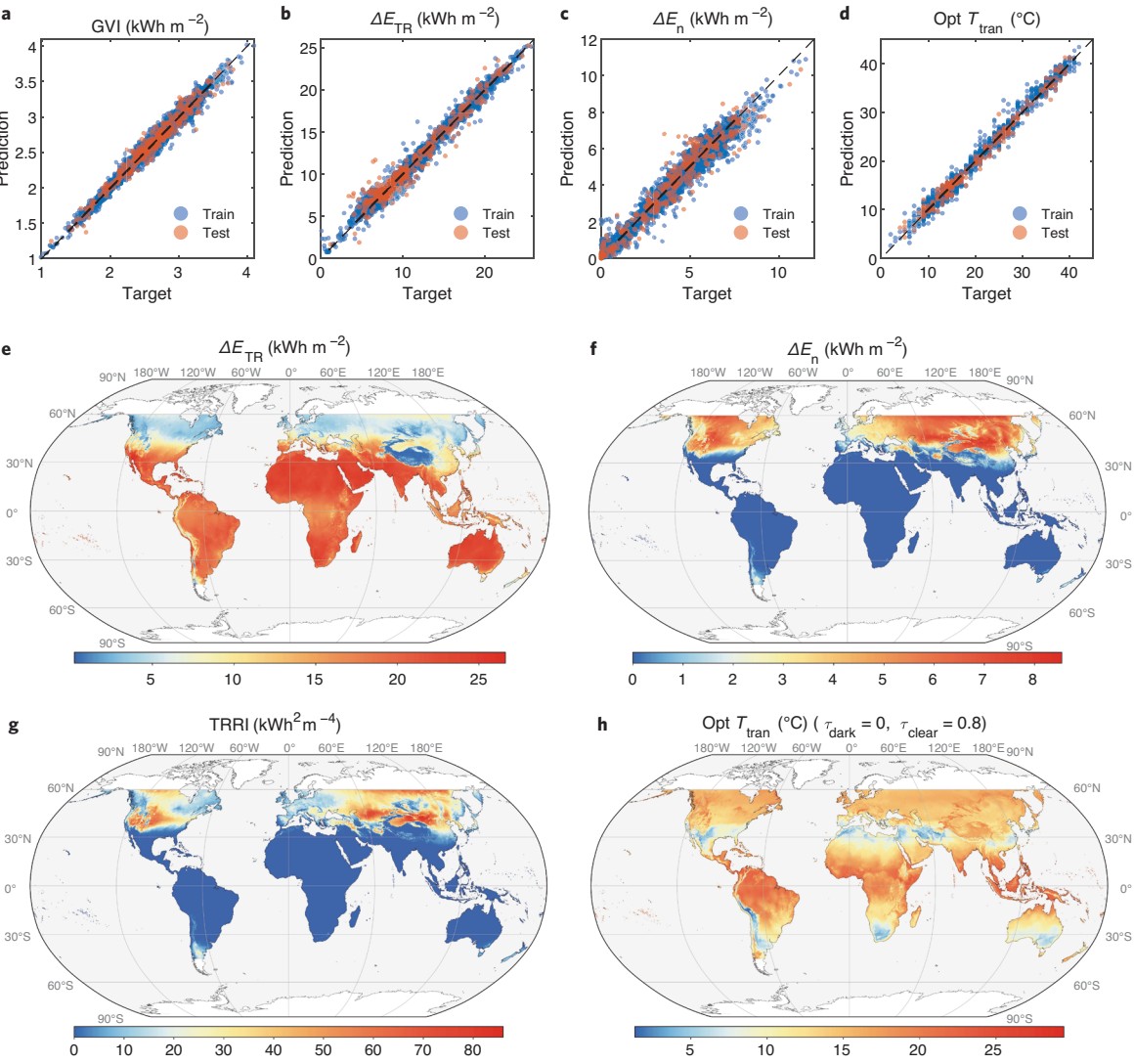

**Fig. 4 | Global prediction of thermo-responsive (TR) window performance.**
**a–d** Training and testing results of artificial neural network (ANN) models for global vertical irradiance (GVI) (**a**), total site energy saving ($\Delta E_{TR}$) (**b**), necessity level ($\Delta E_n$) (**c**) (**b** and **c** are trained by data under optimal conditions), and optimal transition temperature (opt $T_{tran}$) (**d**) (**d** is trained by data under all conditions). **e–h** World

heatmaps of $\Delta E_{TR}$ (**e**), $\Delta E_n$ (**f**), TRRI (**g**) (under optimal conditions), and opt $T_{tran}$ (**h**) (when $\tau_{clear} = 0.8$, and $\tau_{dark} = 0$). Coastlines, country, and state boundaries are provided by the Natural Earth dataset (public domain). Source data are provided as a Source Data file.

windows. We subsequently delve into the analysis of two specific TC materials within North America—$VO_2$ (with $\tau_{dark} = 0.35$, with $\tau_{clear} = 0.52$, and $T_{tran} = 37$ °C as shown in Fig. 5a–c) and hydrogel (with $\tau_{dark} = 0$, $\tau_{clear} = 0.8$, and $T_{tran} = 32$ °C as shown in Fig. 5g–i)—to underscore the composite effect of multi-parameters. Upon modifying $T_{tran}$ from its initial value to room temperature (25 °C), and then to its optimal level, we note a decrease in the positive area of $\Delta E_n$ on the map, alongside an increase in $\Delta E_{TR}$ values, illustrating the direct impact of $T_{tran}$ adjustments on performance metrics. This implies that an improperly selected $T_{tran}$ could negate the need for TR windows entirely (Fig. 5g), whereas improper optical dynamics ($\tau_{dark}$ and $\tau_{clear}$) might lead to more energy consumption than baseline, even with an optimal $T_{tran}$ (Fig. 5c, f). Zooming into three representative states in the US, scatter plots weighted by population density identify the optimal $T_{tran}$ ranges based on $\Delta E_{TR}$ and $\Delta E_n$ (Fig. 5d–f, j–l). It suggests that materials with a broader range of solar modulation could enhance energy savings and amplify the necessity for incorporating TR windows. The regional analysis will enable local manufacturers and vendors to customize and fine-tune the essential characteristics of TR window products.

Based on the results above, we provide an open-source Python tool (https://github.com/LBNL-ETA/PyDynamicWindow) (this link might not be available for reviewers at this moment, but will be published before the paper is online) for researchers to quickly evaluate different TR materials. A world heatmap of desired indices ($\Delta E_{TR}$, $\Delta E_n$, and TRRI) or optimal properties (optimal $T_{tran}$) can be obtained within seconds by simply inputting the fixed properties of TR materials. ANN models with more variables (e.g., building types, window orientations, etc.) and more indicators (e.g., total source energy) will be trained in the future, and new versions will be released upon request. A short demonstration and tutorial video about this open-source tool can be found in Supplementary Video 1.

## Discussions

It's important to understand that the derived values from the ANN models are only valid under specific conditions. For instance, the optimal $T_{tran}$ could differ when the TR layer is applied on the interior window surface rather than the exterior one. The proposed indicators and tools for architectural TR windows offer pathways for material scientists to optimize their TR materials for energy-efficient smart

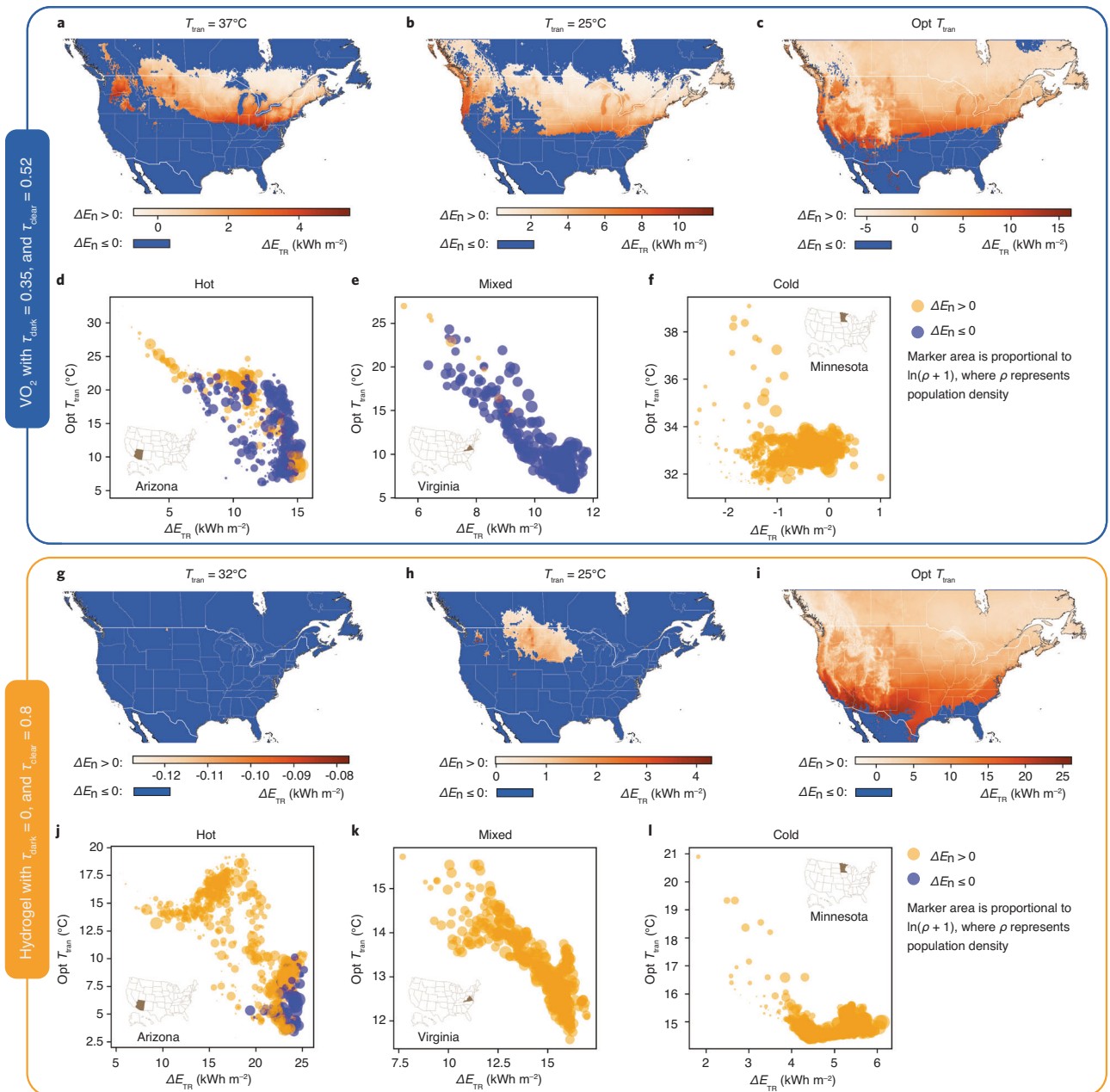

**Fig. 5 | Regional energy performance of thermo-responsive (TR) windows with actual TC materials. a–c** Regional heatmaps of total site energy saving ($\Delta E_{TR}$) when necessity level ($\Delta E_n$) > 0 for Vanadium dioxide (VO$_2$) windows with various transition temperatures ($T_{tran}$). **d–f** Optimal transition temperature (opt $T_{tran}$) and corresponding $\Delta E_{TR}$ for VO$_2$ windows in three representative states in the US. **g–i** Regional heatmaps of $\Delta E_{TR}$ when $\Delta E_n$ > 0 for hydrogel windows with various

$T_{tran}$. **j–l** Opt $T_{tran}$ and corresponding $\Delta E_{TR}$ for hydrogel windows in three representative states in the US Regional heatmaps (**a–c**, **g–i**) are partial maps of North America, covering latitudes from 24.4°N to 55°N and longitudes from 135°W to 50°W. Coastlines, country, and state boundaries are provided by the Natural Earth dataset (public domain). Source data are provided as a Source Data file.

windows. The following steps provide a guideline: (1) Determine if the TR materials are developed for typical office space, based on which ANNs have been trained in this study (no lighting control, TR layer applied on the exterior surface of a clear-clear double-pane window). If not, utilize the preferred EnergyPlus models to regenerate energy-saving results and retrain the ANNs. (2) Strive to achieve the lowest possible $\tau_{dark}$ and highest possible $\tau_{clear}$. (3) Enter $\tau_{dark}$ and $\tau_{clear}$ into the ANN for the optimal $T_{tran}$ and obtain the heatmap and latitude distribution. (4) Tune the $T_{tran}$ of the TR windows to the value or range that benefits the most or specific areas or populations. (5) Input $\tau_{dark}$, $\tau_{clear}$, and $T_{tran}$ into the ANNs for $\Delta E_{TR}$, $\Delta E_n$, and TRRI and obtain global heatmaps. Conversely, we recommend researchers to execute specific

EnergyPlus simulations for a particular building using a given whether file if the installation location of the TR windows is predetermined. We would also like to highlight that this study mainly considered building energy savings of TR windows. Visual comfort is beyond the scope of this study, and while related information can be found in other sources[40], the results may vary depending on the specific context and methodology used. A dark window can effectively reduce glare, but it may compromise the quality of the view. A potential solution for TC windows is to incorporate a partial opening without TC materials, allowing occupants to enjoy a clearer view.

The deployment of applied film retrofit represents a feasible entry point for TR products into the window market, offering a cost-effective

alternative to comprehensive window retrofits in terms of both materials and installation. Additionally, its reversibility presents minimal risk to consumers, allowing for easy removal if the product fails to meet occupant expectations. Positioning TC films on the exterior surface of windows has been shown to yield higher energy savings compared to interior applications. Nonetheless, this approach necessitates the development of durable TC films capable of enduring the rigors of external environmental conditions. Conversely, EC films are typically installed inside due to considerations of protection, ease of maintenance, and the practicalities of connectivity and wiring. Sensitivity analysis indicates a higher optimal $T_{tran}$ for interior installations compared to exterior ones, as films positioned externally are directly exposed to sunlight, leading to a more rapid and significant increase in temperature.

In summary, we systematically analyzed the energy performance of architectural thermo-responsive (TR) dynamic windows under various conditions: differing material properties, window configurations, building and environmental circumstances. We found that optimal dark-state solar transmittance is zero for buildings without lighting control, and slightly above zero for those with lighting control. High clear-state solar transmittance was shown to enhance energy saving, substantially impacting the optimal transition temperature. Other factors, such as the applied window surface, window-to-wall ratio, electrical equipment level, heating and cooling setpoints, etc., also influence the optimal transition temperature. To guide researchers in determining the optimal properties and suitable climates for TR windows, we introduced two metrics: the necessity level and the TR recommendation index. These proposed metrics can also be applied to evaluating other types of dynamic window products. Our study provides compelling evidence that in most of low-latitude tropical areas, TR windows perform equivalently to static windows in terms of energy savings, as reflected by a recommendation index of zero. To assist researchers in optimizing and evaluating TR materials and windows, we offer an open-source mapping tool. This tool identifies climate conditions that are not good applications of TR windows and assists researchers in improving their materials and products for target climates.

## Methods

### Optical measurement and parameterization
We performed optical measurements of thermochromic materials using an edge-heating and temperature control system, which has high repeatability and high resolution of the temperature of the material during the measurement. The experimental setup incorporates a specimen frame to securely clamp heating elements at the edges of the sample under investigation. A series of thermocouples is strategically positioned across the sample's surface to monitor the thermal gradient[41]. The measured results are then used to make the parametric spectral solar transmittance, reflectance and absorptance (Supplementary Fig. 125).

### Simulation batches and corresponding variables
The server used in the study is equipped with an Intel Xeon Platinum 8160 CPU running at 2.10 GHz. The server configuration includes two processors, providing a total of 48 cores and 96 logical processors. The server is equipped with a substantial 256GB RAM. The energy performances of TR windows were evaluated using three simulation batches, starting with a broad resolution and a wide variable range, then progressing to a finer resolution within a narrowed variable range according to statistical analysis on the previous batch. This approach ensures a balance between the variable resolution and computational load. Details about the three simulation batches were summarized in Supplementary Table 2. The initial simulation batch focused on three types of variables that are crucial in achieving building energy savings, including TR properties ($\tau_{clear}$, $\tau_{dark}$, and $T_{tran}$), window configurations

(glazing configuration, and applied surface), and building configurations (location, window orientation, and lighting control). Multi-pane windows exhibit low $U$-values, which improve the thermal insulation. A low-emissivity (low-e) coating on glass can regulate solar heat gain coefficient (SHGC), which measures the capability of a window collecting (high SHGC) or blocking (low SHGC) the heat gain from the sun. TR materials are applied on five baseline glazing configurations, i.e., single clear, double clear, double low-e, triple low-e #1 (low SHGC), triple low-e #2 (high SHGC). Theoretically, the TR layer can be applied to any surface of a single or multi-glazing system, among which the exterior surface (Surface #1) and interior surface (Surface #2, 4, or 6 for single, double, or triple glazed window, respectively) were used due to retrofit purpose and performance representativity (Fig. 1b). Sixteen US climate zones and representative cities used in this study are listed in Supplementary Table 3[35]. Four window orientations (south, east, west, and north) and both lighting control schemes (LC No, and Yes) were considered. In the second simulation batch, variable values for $T_{tran}$ were increased from 8 to 43 with finer resolution (1 °C). Meanwhile, other variable values were reduced to one applied surface (Surface #1), and three window orientations (south, east, and west). The third simulation batch, compared with the second one, increased the weather conditions from 16 to 2226 (marked in a world map in Supplementary Fig. 75), and further fixed window orientation (equator-facing), glazing configuration (double clear), and lighting control scheme (LC No).

### Reference building models and energy simulation
EnergyPlus (version 9.2), an open-source building energy simulation program, is used to predict the annual energy performance of reference buildings with TR and baseline windows. In EnergyPlus, a thermochromic window is represented through a Construction object, linked to a specialized layer defined by a Window-Material:GlazingGroup:Thermochromic object. This Window-Material:GlazingGroup:Thermochromic object, in turn, refers to an array of WindowMaterial:Glazing objects, each one corresponding to the distinct specification temperature of the thermochromic layer. The office space used in the parametric study is adapted from the US DOE reference model for a medium-sized office building (Supplementary Fig. 49)[36]. The HVAC system in this model utilizes a two-speed direct expansion (DX) compressor and fan for cooling and a fuel heating coil for heating. The building models of medium-size offices and mid-rise apartments are also used for the supplementary study (Supplementary Figs. 66, 67)[36].

### Statistical analysis of first simulation batch
Supplementary Figs. 1–16 shows the statistical analysis of 1920 simulation cases for each window orientation and each representative city in the first batch. Results reveal that the maximum $\Delta E_{total}$ is attained when TR film is applied to the exterior surface (Surface #1), and the glazing configuration is double clear. According to those results, further statistical analysis focused on LC No (Supplementary Figs. 17–32) and LC Yes (Supplementary Figs. 33–48) by also fixing the TR applied surface (Surface #1) and glazing configuration (double clear).

### World map data for inputs
Solar radiation data are obtained from the Global Solar Atlas 2.0, a free, web-based application developed and operated by the company Solargis s.r.o. on behalf of the World Bank Group, utilizing Solargis data, with funding provided by the Energy Sector Management Assistance Program (ESMAP). For additional information: https://globalsolaratlas.info[37].

Global climate data for minimum, mean, and maximum temperature are obtained from https://www.worldclim.org/data/worldclim21.html[38].

Global data of population density is obtained from https://neo.gsfc.nasa.gov/view.php?datasetId=SEDAC_POP. Data are produced by the NASA Earth Observations team based on data provided by the Socioeconomic Data and Applications Center (SEDAC), Columbia University[39].

## Artificial neural networks

Multi-layer perceptron (MLP) regression, a type of artificial neural network, is used to analyze our data. MLP is chosen due to its ability to learn complex mappings from inputs to outputs, which is particularly effective for modeling non-linear relationships. The structure of our MLP consists of an input layer, multiple hidden layers, and an output layer. Our dataset is split into a training set and a test set in an 85–15% ratio, respectively. The MLP model is trained using the training set via a back-propagation algorithm, which iteratively adjusts the weights of the model to minimize the error - the difference between the predicted and actual values.

To find the optimal hyperparameters, such as the number of hidden layers, the number of neurons in each hidden layer, the activation function, etc., we employ a hybrid search strategy using three search methods: Optuna, random search, and grid search. L1 and L2 regularization techniques are used to avoid overfitting, providing a balance between the model's complexity and its ability to generalize from the training data. The optimal hidden layer sizes for GVI, $\Delta E_{TR}$, and $\Delta E_n$ are (168, 102, 144, 97, 141, 184, 113), (154, 124, 125, 157, 89, 129), and (161, 167, 165, 198) respectively. For all models, the 'ReLU' activation function proved optimal. All MLP regression models are implemented using the Python package scikit-learn, selected for its superior performance and the flexibility that Python provides in data analysis and modeling.

## Stepwise model validation

Though it is challenging to directly validate our proposed models, it is feasible to validate the model step by step. We analyzed the accuracy of the EnergyPlus software, the built-in thermochromic model, ANNs and weather inputs, respectively. The credibility has been demonstrated by results from both previous literatures and our study. Detailed Credibility Analysis can be found in Supplementary Note 1.

## Data availability

The original data that support the findings of this study are available from the corresponding author upon request. The source data underlying Figs. 1–5 are provided as a Source Data file. Source data are provided with this paper.

## Code availability

The code for generating heatmaps of indicators for TC windows is available on GitHub (https://github.com/LBNL-ETA/PyDynamicWindow). Other codes for this study can be make available upon request.

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

## Acknowledgements
This work was supported by the Assistant Secretary for Energy Efficiency and Renewable Energy, Building Technologies Program, of the US Department of Energy under Contract no. DE-AC02-05CH11231. This work was also supported by the US Department of Energy Building Technologies Office (BTO) Buildings Energy Efficiency Frontiers & Innovation Technologies (BENEFIT) (D.C.C.). The authors sincerely thank Dr. Jie Li at Argonne National Laboratory for providing $VO_2$ samples. The authors would like to express sincere appreciation to Robin Mitchell, Stephen Czarnecki, and Taoning Wang, for their assistance in publishing the Python tool.

## Author contributions
Y.G. led the efforts on model developing, simulation, analysis, and visualization. J.C.J. contributed to the optical measurements and parameterization. Y.G., J.C.J., and D.C.C. formulated the original concept of necessity level and recommendation index. S.V. contributed to the development of artificial neural network models. Y.G., J.C.J., and T.H. contributed to preparation of the manuscript, and all authors read and commented on the manuscript. D.C.C. directed and managed the overall project.

## Competing interests
The authors declare no competing interests.
