## [Transparent Peer Review file · Nature Communications]

Global and regional perspectives on optimizing thermo-responsive dynamic windows for energy-efficient buildings

Corresponding Author: Dr Yuan Gao

Version 0:

Reviewer comments:

Reviewer #3

(Remarks to the Author)

This study presents the energy saving potential of TR windows, including electrochromic (EC) windows and windows with thermochromic (TC) materials, and the correlations between climate suitability and intrinsic material properties on a worldwide scale. The described study examines over one hundred TC and EC samples utilizing computer simulations and covering a range of windows and building configurations and weather conditions.

Key results

The investigation resulted in the conclusion that TR windows can provide maximum energy overall savings (lighting, cooling/heating) in mixed-climate regions (needing heating and cooling) as compared to tropical areas (requiring only cooling). An open-source tool for researchers or building professionals to identify locations where TR windows can function efficiently has been developed and provided.

Significance

The significance of the study is high. The amount of work invested in the study is considerable and the tool that has resulted can be of wide use to academics and professionals.

Validity

The simulation process appears to be detailed and taking into consideration the maximum number of parameters (climate, space, orientation, materials). The simulation results and the tool provided have the potential of working as guides for the use of TR windows. However, the study does not include the validation of the simulated performance data compared against existing experimental data from all over the world.

This would be a more general criticism of the paper, which is almost neglecting to discuss and present previous experimental results on the energy performance of TR windows.

Suggestions:

1. As mentioned previously, a paragraph reviewing the relevant experimental studies and validating the simulated results against experimental data is missing and needs to be added.
2. Tables with basic definitions are needed, both for the existing indices (τ_{clear} , τ_{dark} , T_{tran}) as well as the proposed ones (ΔETR , E_{dark} , etc).
3. A clearer structure of the paper is needed, with more robust titles that allow the reader to understand which part of the paper they read at any time. Also, the discussion section reads more as "tool manual" than actual discussion, as it only refers to the tool and not on the simulations and actual research findings.

(Remarks on code availability)

Reviewer #4

(Remarks to the Author)

Review of NCOMMS-24-18413-T, entitled "Global and regional perspectives on optimizing thermo-responsive 2 dynamic windows for energy-efficient buildings" submitted to Springer Nature to be considered for publication

To the Authors

The paper proposes a summary of an enormous investigation concerning the thermal performance of thermo-responsive dynamic windows in managing solar gains, and thus in improving thermal behavior and energy demands of buildings. The main novelty of the study is the global-scale analysis, starting from the U.S. conditions, carried out by developing artificial neural network models for suitably predict performance indicators of TR windows, also in terms of world heatmaps indicators for TC windows, and thus allowing a powerful tool for simulating the thermal performance of TR windows in many geographical sites. More than two thousand global weather conditions have been used for the training of ANNs. Also, the authors provide software to optimize TR materials, consisting of an open-source mapping tool. About the results of the study, these are very wide and interesting, following a large analysis of outcomes from more than 2.8 million simulations (considering all main technologies and materials, and thus peculiarities), and these revealed the usefulness of thermo-responsive windows mainly for buildings characterized by both significant energy demands for space heating and space cooling. It should be noted that, for reason of brevity, merely few results and good discussion are provided in the main manuscript, while much other info, data, figures and tables are reported in the supplementary materials really worthy and meaningful. My general opinion about this manuscript is surely favorable. It concerns an important topic, a useful arrow to improve, in future years, thermal and energy performance of both new and existing buildings, by proposing a method and a technology that can be applied worldwide, without limitation of architectural technology or construction techniques, but suitable (or not) only based on technical and thermodynamic considerations, for which the proposed methods (and tools) are an important support to use and thus to decide. That's why I am favorable to this study: it is global and purpose, besides some results, also a tool to apply and check such technologies in different locations and by optimizing the configuration.

I have some minor suggestions for the authors:

- At line 29, it could be worthy to cite the impacts of buildings on the energy demands and mandatory policies (if any) for improving the energy performance of the construction sector in different world areas, e.g., Europe, North and South America, Asia, Oceania and Africa. This may be useful to understand the impacts of buildings on energy demands at a global scale. Indeed, the global scale is the "spatial domain" of this investigation.
- The introduction of the paper: some lumped references do not allow the specific contribution of each cited paper. For instance, check 8-19 (line 34): the sentence used to cite such papers is too general.
- The introduction describes the main properties and features of TC windows and thus the clear-state solar transmittance, the dark-state solar transmittance, the transition temperature, the available thermochromic materials and so on. There are, on the other hand, no summary information about the real use of TC windows in buildings. Is this technology spread, and used, in the sector? How much and in which countries it is common?
- At line 92, it is written that a comprehensive guideline is provided. Please, can you specify in which section of the paper is it?
- Phoenix, Baltimore, and Minneapolis, according to the authors, are cities representing typical climates (hot, mixed, and cold). Please, may you provide more data? For instance, the Köppen-Geiger classification, HDD and CDD and so on?
- Many symbols, parameters, and acronyms are used. Many times, I have searched for the meaning of a symbol, without an immediate understanding. Please, provide a full nomenclature, with all parameters and SI units.
- Figure 3 may be more readable if readers can immediately understand the climate of such locations. Could you provide a map with HDD and CDD or similar indicators of the coldness and warmth of the U.S. locations?
- At the world level (figure 4), more or less the climates are clear, so the previous request is not so mandatory.
- I suggest rephrasing at line 353 (Visual comfort is not within the scope of the study and can be found elsewhere). For sure it can be found elsewhere but probably the results will be different. So, please rephrase, now it seems that paper 40 is connected to this one.
- Just a curiosity concerning Figure 4, last picture (down and right, 4h). How do the authors explain that Mediterranean countries have so low optimal transition temperature (around 5 °C) and this is valid mainly for the African coasts of the Mediterranean and not for the European coast (where higher optimal T trans occur)?
- Supplemental materials and explanations at the end of the paper are useful and meaningful, well-done.
- Please, given that the whole study is based on simulation, provide a validation of the main energy results and outcomes of the initial 3 cities (Phoenix, Baltimore, and Minneapolis).
- Add limitations and future perspectives. For instance, the next study may be based on validation and comparison against experimental results.

Finally, in my opinion, this is a comprehensive paper, novel, providing interesting results and giving contribution to the research community involved in the field. I recommend a minor revision.

(Remarks on code availability)

I checked all submitted materials, including ones on Git Hub, but I did not run the code.

Version 1:

Reviewer comments:

Reviewer #1

(Remarks to the Author)

I would like to thank the authors for their thorough review of the manuscript and the detailed response to my comments. I believe that the manuscript has improved significantly since it was first submitted.

I insist that this is a significant piece of work that should be published, due to its usefulness to the industry and academia. However, I will also need to insist that any paper that is based on simulations needs to validate its results. I realise the difficulties of reproducing the buildings, or having full-sized samples. I also appreciate the fact that the tool used (EnergyPlus), as well as the rest of the components are all individually validated and/or commonly used. The validity of each of the components, however, does not guarantee the validity of the final product. All models are wrong, but the non-validated ones can be more wrong and potentially not useful.

It is my belief that the authors should either find a way to experimentally validate their results or propose another methodology to prove that the tool provides accurate results.

(Remarks on code availability)

Reviewer #2

(Remarks to the Author)

Review of NCOMMS-24-18413-T, entitled "Global and regional perspectives on optimizing thermo-responsive 2 dynamic windows for energy-efficient buildings" submitted after revision to be considered for publication in Springer Nature Communications.

To the Authors

An extensive investigation is proposed to review the thermal performance of thermo-responsive dynamic windows, the management of solar gains, and thus, the improvement of thermal behavior and energy demands of buildings. A novel global-scale analysis is proposed, even by developing artificial neural network models for suitably predict performance indicators of TR windows; thus, a powerful tool for simulating the thermal performance of TR windows in many geographical sites is developed, and the use of the framework, by applying millions of simulations, provided many interesting results and a worthy discussion. I have had a very positive opinion already concerning the original submission. The method is sound, the investigation is novel, conclusions are supported by data. Really, I asked for some deeper info and comments, some better explanations (for instance, about climates, HDD, and CDD), some additional discussion, avoiding of lumped references, comments about validations, limits of the present investigation (by commenting about these, if any). The authors replied carefully, adding both new parts in the manuscript and the supplementary materials. They answered my comments and curiosities, and everything was clear, careful, and meaningful. My opinion is that the current version is wide, large, interesting, and worthy to be published in this prestigious journal. Accept.

(Remarks on code availability)

made.

Point-by-point actions in response to the Editors' and Reviewers' comments

Manuscript # NCOMMS-24-18413-T

We sincerely thank the editors and reviewers for their valuable suggestions, which have enabled us to improve the manuscript. Following are the detailed actions taken in light of editors' and reviewers' comments:

Reviewer #1 (Remarks to the Author):

This study presents the energy saving potential of TR windows, including electrochromic (EC) windows and windows with thermochromic (TC) materials, and the correlations between climate suitability and intrinsic material properties on a worldwide scale. The described study examines over one hundred TC and EC samples utilizing computer simulations and covering a range of windows and building configurations and weather conditions.

Key results

The investigation resulted in the conclusion that TR windows can provide maximum energy overall savings (lighting, cooling/heating) in mixed-climate regions (needing heating and cooling) as compared to tropical areas (requiring only cooling). An open-source tool for researchers or building professionals to identify locations where TR windows can function efficiently has been developed and provided.

Significance

The significance of the study is high. The amount of work invested in the study is considerable and the tool that has resulted can be of wide use to academics and professionals.

Response: We appreciate the reviewer's positive feedback on our work. This study aims to provide fresh perspectives for scientists, engineers, architects, and policymakers, aligning with the diverse target audience of Nature Communications.

Validity

The simulation process appears to be detailed and taking into consideration the maximum number of parameters (climate, space, orientation, materials). The simulation results and the tool provided have the potential of working as guides for the use of TR windows. However, the study does not include the validation of the simulated performance data compared against existing experimental data from all over the world.

This would be a more general criticism of the paper, which is almost neglecting to discuss and present previous experimental results on the energy performance of TR windows.

Response: First and foremost, we would like to extend our gratitude to the reviewer for the rigorous academic standards and for pointing out the shortcomings of our study.

In most material research, it is highly challenging to fabricate full-sized TR windows and conduct experiments in actual buildings, especially for labs that can only produce small-size samples. Instead, most studies experimented their samples in small-scale chambers (see Table 2 in *Cell Reports Physical Science* 4.5 (2023)) or evaluating the energy performance of materials in buildings through simulations. Only a few studies have employed full-sized TR windows in real buildings for field experiments (summarized in Response Table 1).

A publication in 2013 introduced an experimental work on large-area polymer thermochromic (TC) laminated windows in a full-scale testbed office in our lab [*Solar Energy Materials and Solar Cells* 116 (2013): 14-26]. Several measured parameters, such as transmitted solar radiation, incident vertical irradiance, and outdoor dry-bulb temperature, were compared against the predicted values. EnergyPlus was used to calculate the annual energy performance for Chicago and Houston. However, the thermochromic module in EnergyPlus was not validated in this study. Both empirical and simulation data were utilized to demonstrate that the ideal critical switching temperature for TC windows should be determined by the zone heat balance rather than the ambient air temperature. Another demonstration work was also reported in 2013 by our lab regarding the energy performance of electrochromic and thermochromic windows in a federal building in Denver [GSA report (2013)]. The study on thermochromic windows involved several key measurements to evaluate their performance, including glazing temperature, vertical irradiance and illuminance. Environmental factors like incident solar radiation, outdoor air temperature, wind, and indoor air temperature were also monitored to understand their influence on the window's switching behavior. EnergyPlus simulation was conducted for annual energy calculation and was not validated by measured data. In 2013 and 2015, two publications reported the demonstration and simulation of VO₂ single glazing [*Solar energy materials and solar cells* 117 (2013): 168-173] and double glazing [*Solar Energy* 120 (2015): 55-64] in a full-scale room in Hefei, China, respectively by the same research team. The simulated cooling load was validated by the measured data. However, the simulation tool is BuildingEnergy software, which is more user-friendly, but less customizable compared with EnergyPlus. They concluded that the TC windows were suitable for hot climates rather than cold climates. VO₂ double window consumes approximately 11.1% less cooling energy than that with an ordinary double window. Another paper published in 2015 reported testing thermotropic glazing and a triple glazing unit in a full-scale outdoor test cell to evaluate their thermal and optical performance [*Proceedings of building simulation* (2015)]. Measurements were taken over several days under varying solar radiation and temperature conditions, focusing on internal glass surface temperature, transmitted solar radiation, and heat flux. The experimental data were then compared with simulation results from EnergyPlus to assess the accuracy of the built-in thermochromic model. Results showed that the errors of simulated internal glazing surface temperature, transmitted solar radiation, and heat

flux stayed in acceptable range, proving the accuracy of the built-in thermochromic model in EnergyPlus. This study will be introduced in detail later in the Credibility Analysis Section (B).

Response Table 1 | Summary of experimental study on full-size TC windows in real building environment

Publishing year	TC material	Building	Location	Simulation software	Validation	Reference
2013	Polymer	Full-scale, south-facing, conditioned testbed office	Berkeley, California, USA	EnergyPlus	No	Solar Energy Materials and Solar Cells 116 (2013): 14-26
2013	Not mentioned	9,500-ft ² perimeter zone of a Federal office building	Denver Federal Center, Colorado, USA	EnergyPlus	No	GSA report (2013)
2013	VO ₂	2.9 x 1.8 x 1.8 m ³	Hefei, China	BuildingEnergy	Yes	Solar energy materials and solar cells 117 (2013): 168-173
2015	VO ₂	2.9 x 1.8 x 1.8 m ³	Hefei, China	BuildingEnergy	Yes	Solar Energy 120 (2015): 55-64
2015	Technology based on a coreshell particle suspension	1.6 x 3.6 x 2.5 m ³ TWINS outdoor test cell	Torino, Italy	EnergyPlus	Yes	Proceedings of building simulation (2015)

Please allow us to explain why it is extremely challenging to directly validate the results of this study. Due to the extensive range of TR window parameters and global climate conditions covered in this research, it is only feasible to validate one or a few of these variables with real-world experiments. Moreover, the simulations in this study utilize reference buildings from the U.S. Department of Energy’s (DOE) prototype building models, chosen for their representativeness. The window data in this study were obtained from measurements of small-size samples. In practice, constructing buildings identical to the standard model and manufacturing windows identical to the sample on a large scale are both time-consuming and costly. Although direct validation is difficult, the reliability of this work can still be indirectly verified by analyzing the credibility of each component of this work.

Credibility Analysis

(A) EnergyPlus

This research primarily utilized EnergyPlus, a whole building energy simulation program. Its development is funded by the U.S. DOE Building Technologies Office (BTO). EnergyPlus has undergone extensive validation and verification processes [Testing and Validation, EnergyPlus]. It has been tested against empirical data from real buildings and benchmarked against other established simulation tools [IEA SHC Task 34/Annex 43]. It has been widely adopted in the industry and academia, with numerous peer-reviewed studies attesting to its accuracy and robustness [research articles related to EnergyPlus].

A previous study validated EnergyPlus by comparing simulation results against measured data from real-world buildings and controlled test environments [Building Simulation 2019, Vol. 16, IBPSA, 2019]. It is an empirical validation project conducted by multiple national laboratories, including Oak Ridge National Laboratory (ORNL). The validation utilized ORNL's Flexible Research Platform, a small office building with detailed monitoring systems. The project generated extensive empirical data sets, including cooling energy consumption, fan energy consumption, and zone temperatures, under various test conditions. The accuracy of EnergyPlus was quantified using metrics such as Normalized Mean Bias Error (NMBE) and Coefficient of Variation of the Root Mean Square Error (CV(RMSE)),

$$NMBE = \frac{1}{\bar{M}} \frac{\sum_{i=1}^n (M_i - S_i)}{n} \times 100\%$$
$$CV(RMSE) = \frac{1}{\bar{M}} \sqrt{\frac{\sum_{i=1}^n (M_i - S_i)^2}{n}} \times 100\%$$

where M , S , and n represent the measurement, simulation, and the number of data, respectively. Upper bar refers to the average. The hourly NMBE and CV(RMSE) were less than 2.6% and 5.9%, respectively, indicating that simulation and experimental energy consumption are well matched.

(B) Built-in thermochromic module

Furthermore, the thermochromic module has also been verified by comparing with experimental data in a previous study [Proceedings of building simulation (2015)]. In this study, simulation results using EnergyPlus and the built-in thermochromic module were compared with the experimental results measured in a full-scale outdoor test facility (1.6 m x 3.6 m x 2.5 m) in Torino, Italy. The experimental data was collected using the side-by-side test cell facility equipped with a triple glazing unit (TGU) and a triple glazing unit with thermotropic (TT) glazing on the external side (TT+TGU). The data for comparison was taken from April 12th to April 15th, 2013, which included medium to high vertical solar radiation and temperature variations. Response Fig. 1

shows the results of the reference TGU, with good agreement between simulation and experimental data in general. Simulated and measured parameters include internal surface temperature of the glazing ($T_{glass,in}$), transmitted solar radiation (G_{in}), and heat flux (radiative longwave and convective) on the internal surface of the glazing (HF_{lw}). A two-hour delay was observed in measured temperatures compared to simulation due to EnergyPlus not accounting for the thermal mass of the glazing. This issue could be solved by using equivalent models as demonstrated in our previous work [*Applied Energy* 301 (2021): 117467]. Please note that here our parametric study on TC windows doesn't take the thermal mass of the glazing into account. A peak difference in the heat flux in the afternoon was also observed, likely due to the heat flow meter sensor overheating from direct solar radiation [*Proceedings of building simulation* (2015)].

Response Fig. 1 | Simulation and experimental results of the reference TGU. a. Transmitted solar radiation (left y axis), and internal surface temperature of the glazing (right y axis). b. Heat flux on the internal surface of the glazing. Figure adapted from [*Proceedings of building simulation* (2015)].

To quantitatively evaluate the simulation performance, the errors between simulation and experimental data were defined as below:

Mean Bias Error (MBE):

$$MBE = \frac{1}{n} \sum_{i=1}^n (X_{mod} - X_{exp})$$

Root Mean Square Error (RMSE):

$$RMSE = \sqrt{\frac{1}{n} \sum_{i=1}^n (X_{mod} - X_{exp})^2}$$

Percentage Root Mean Square Error (PRMSE):

$$PRMSE = \sqrt{\frac{1}{n} \sum_{i=1}^n \left(\frac{X_{mod} - X_{exp}}{X_{exp}} \right)^2}$$

where n is the number of measurements.

The results are listed in Response Table 2. The errors remain within an acceptable range, demonstrating that the simulation results for the reference room using EnergyPlus are reliable.

Response Table 2 | Simulation errors compared with measured results in the cell with TGU [*Proceedings of building simulation (2015)*]

	MBE	RMSE	PRMSE
$T_{glass,in}$ (°C)	-0.5	1.5	5.1%
G_{in} (W m ⁻²)	-0.6	11.5	-
HF_{lw} (W m ⁻²)	-6.3	13.7	-

In Response Fig. 2, “TT+TGU_E+” indicates the simulation results of triple glazing unit with thermotropic glazing using the built-in thermochromic model. “EMS – E+” indicates an alternative EMS (Energy Management System) EnergyPlus model, which will not be considered in this discussion. The TT+TGU simulation showed good alignment with measured solar radiation data but exhibited a 2-hour delay and peak temperature differences of 3-4°C during peak solar radiation hours. Additionally, heat flux discrepancies of 10-15 W/m² were noted. These discrepancies, primarily due to increased solar reflectance of the TT glazing in experimental conditions, needed a model calibration to better match the experimental data. The calibration involved increasing the solar and luminous reflectance by a constant factor, which significantly improved model accuracy (denoted with “_mod” in Response Fig. 2). After calibration, the MBE for $T_{glass,in}$ was reduced to -0.01°C, with a RMSE of 1.38°C and a PRMSE of 5%. Similarly, for the HF_{lw} , the MBE was reduced to -0.84 W/m², and the RMSE to 10.31 W/m². These values indicate that the calibrated built-in TC model in EnergyPlus is reliable for predicting the performance of TC glazing technologies.

Response Fig. 2 | Simulation and experimental results of TT+TGU. a. Internal surface temperature of the glazing. b. Heat flux on the internal surface of the glazing. c. Transmitted solar radiation. d. Table of simulation performance of different methods. Figure adapted from [Proceedings of building simulation (2015)].

(C) ANNs

Our study reported multiple different artificial neural networks (ANNs), each trained and tested using different inputs and outputs. Here we select several representative ANNs and their performance, which are displayed in Response Fig. 3. The training of each neural network was based on the mean squared error (MSE), defined as

$$MSE = \frac{1}{n} \sum_{i=1}^n (S_i - M_i)^2,$$

where S and M represent the predicted and actual values, respectively. Based on MSE values, we calculated other error metrics, which are listed in Response Table 3. Different ANN structures exhibit various performances as shown in Response Fig. 3. Generally, the more effective data can be used to train the ANN, the better its performance becomes. It is noteworthy that due to the inherent randomness in neural network training, the same ANN structure may yield different MSE values across different trainings. In this study, we selected the best-performing results from multiple trainings. Although errors are inevitable, the linear regression between the actual and predicted values of the ANNs is obvious, enabling the ANNs to provide valuable guidance for the

design of the TR window. From the error metrics, we can conclude that the errors of the ANNs in this study stay within an acceptable range (Response Table 3).

Supplementary Figure 74a

Figure 4a

Supplementary Figure 122a

Supplementary Figure 123b

Supplementary Figure 122b

Supplementary Figure 124b

Supplementary Figure 118c

Supplementary Figure 121b

Response Fig. 3 | Four selected ANNs and the performance of training and testing. Figures are adapted from manuscript and SI.

Response Table 3 | Testing performance of four selected ANNs in this study

Output of ANN	MSE	RMSE	CV(RMSE)
GVI	0.0080	0.0894	3.4%
ΔE_{TR} (trained by all conditions)	0.0160	0.1265	1.3%
ΔE_n (trained by all conditions)	0.0144	0.1200	3.8%
Optimal T_{tran} (trained by the optimal conditions)	2.4458	1.5639	7.9%

(D) Input data of solar radiation and air temperature

Solar radiation

Solar radiation inputs of this study include direct normal irradiation (DNI), global horizontal irradiation (GHI), and diffuse horizontal irradiation (DHI), which are obtained from the Solargis solar radiation model [Global Solar Atlas]. In a validation report, the accuracy of solar radiation data was calculated through the comparison with ground-data from the reference stations [Global Solar Atlas 2.0 : Validation Report]. In this report, the solar radiation data has been validated at 228 public sites worldwide. The bias (in percent) of GHI and DNI are shown in Response Fig. 4 and 5, respectively. The mean bias of GHI and DNI for all sites are 0.3% and 2.2%, respectively. Besides the validation in this report, a list of independent validation studies can also be found in Section 6 of this report. The Solargis solar radiation model has been proved reliable and the solar radiation data used in this study falls within an acceptable error range.

Response Fig. 4 | GHI bias on the background of climate zones (values in percent). Climatic classes: A – tropical; B – arid; C – temperate; D – cold; E – polar. Figure adapted from [Global Solar Atlas 2.0 : Validation Report].

Response Fig. 5 | DNI bias on the background of climate zones (values in percent). Figure adapted from [Global Solar Atlas 2.0 : Validation Report].

Air temperature

Global air temperature input of this study include minimum, mean, and maximum temperature, which are obtained from WorldClim 2 database [*International journal of climatology* **37**, 4302–4315 (2017)]. The accuracy of the dataset was demonstrated by comparing with station data. RMSE values of minimum, maximum, and average temperature are shown in Response Fig. 6a-c. All temperature variables exhibited a global correlation coefficient of 0.99 or higher between estimated and observed values, and an average RMSE ranging from 1.1 to 1.4°C (Response Fig. 6d). The accuracy is deemed sufficient to predict the trend of building energy performance in a global scale.

Response Fig. 6 | a-c. Spatially aggregated RMSE values of minimum (a), maximum (b), and average (c) temperature. d. Table of global cross-validation statistics for temperature models. Figures adapted from [*International journal of climatology* **37**, 4302–4315 (2017)].

In summary, the accuracy of the EnergyPlus software, the built-in thermochromic model, ANNs and weather inputs are analyzed here. The performance of each component in our proposed models are summarized in Response Fig. 7. As George E. P. Box. said, "All models are wrong, but some are useful." We demonstrated that our models effectively predicted the trend of building energy saving and necessity level of using TR windows.

Response Fig. 7 | Summary of accuracy performance of each component in the proposed models of thermo-responsive dynamic windows.

Suggestions:

1. As mentioned previously, a paragraph reviewing the relevant experimental studies and validating the simulated results against experimental data is missing and needs to be added.

We have now added a paragraph of stepwise model validation in the Method, and a note of credibility analysis in the supplementary information.

Methods: Stepwise model validation

Though it is challenging to directly validate our proposed models, it is feasible to validate the model step by step. We analyzed the accuracy of the EnergyPlus software, the built-in thermochromic model, ANNs and weather inputs, respectively. The credibility has been demonstrated by results from both previous literatures and our study. Detailed Credibility Analysis can be found in Supplementary Note 1.

Supplementary Note 1: Credibility Analysis

(Please see the previous response, which will not be repeated here)

2. Tables with basic definitions are needed, both for the existing indices (τ_{clear} , τ_{dark} , T_{tran}) as well as the proposed ones (ΔE_{TR} , E_{dark} , etc).

We appreciate the reviewer's suggestion. A full nomenclature is now added the supplementary information.

Nomenclature

Abbreviation [unit]	Explanation
τ_{clear} [-]	Clear-state solar transmittance
τ_{dark} [-]	Dark-state solar transmittance
T_{tran} [°C]	Transition temperature
$\Delta\tau_{sol}$ [-]	Solar modulation, i.e., $\tau_{dark} - \tau_{clear}$
ΔE_{total} [kWh m ⁻²]	Total site energy saving per conditioned building (floor) area
E_{total} [kWh m ⁻²]	Total site energy per conditioned building (floor) area
ΔE [kWh m ⁻²]	Energy saving per conditioned building (floor) area
ΔE_{TR} [kWh m ⁻²]	Total site energy saving per conditioned building (floor) area of buildings using TR windows
E_{TR} [kWh m ⁻²]	Total site energy per conditioned building (floor) area of buildings using TR windows
E_{dark} [kWh m ⁻²]	E_{total} by constantly dark (low T_{tran}) TR windows
E_{clear} [kWh m ⁻²]	E_{total} by constantly clear (high T_{tran}) TR windows
ΔE_{dark} [kWh m ⁻²]	ΔE_{total} by constantly dark (low T_{tran}) TR windows
ΔE_{clear} [kWh m ⁻²]	ΔE_{total} by constantly clear (high T_{tran}) TR windows
E_{static} [kWh m ⁻²]	The smaller between E_{dark} and E_{clear} , i.e., $\min(E_{dark}, E_{clear})$
ΔE_n [kWh m ⁻²]	The necessity level of using TR windows compared to static windows, defined as $E_{static} - E_{TR}$
ReLU	Rectified Linear Unit, a piecewise linear function that outputs the input directly if it is positive, otherwise, outputs zero.
TRRI [kWh ² m ⁻⁴]	$ReLU(\Delta E_{TR}) \times ReLU(\Delta E_n)$
DNI [W m ⁻²]	Direct normal irradiation. When it comes daily total DNI, the unit is [kWh m ⁻²]
GHI [W m ⁻²]	Global horizontal irradiation. When it comes daily total GHI, the unit is [kWh m ⁻²]
DHI [W m ⁻²]	Diffuse horizontal irradiation. When it comes daily total DHI, the unit is [kWh m ⁻²]
GVI [W m ⁻²]	Global vertical irradiance. When it comes daily total GVI, the unit is [kWh m ⁻²]
Lat [°]	Latitude
T_{avg} [°C]	Average air temperature
T_{min} [°C]	Minimum air temperature
T_{max} [°C]	Maximum air temperature
TR	Thermo-responsive
CO ₂	Carbon dioxide
HVAC	Heating, ventilation, and air conditioning
TC	Thermochromic
VO ₂	Vanadium dioxide
NIR	Near-infrared
PNIPAM	Poly(N-isopropylacrylamide)
EC	Electrochromic

WO ₃	Tungsten oxide
MoO ₃	Molybdenum oxide
V ₂ O ₅	Vanadium oxide
RME	Reversible metal electrodeposition
SPD	Suspended particle device
PDLC	Polymer dispersed liquid crystal
LC	Lighting control
DOE	Department of Energy
ANN	Artificial neural network
low-e	Low-emissivity
SHGC	Solar heat gain coefficient
DX	Direct expansion
MLP	Multi-Layer Perceptron
BTO	Building Technologies Office
BENEFIT	Buildings Energy Efficiency Frontiers & Innovation Technologies

3. A clearer structure of the paper is needed, with more robust titles that allow the reader to understand which part of the paper they read at any time. Also, the discussion section reads more as “tool manual” than actual discussion, as it only refers to the tool and not on the simulations and actual research findings.

We appreciate the reviewer’s valuable suggestions. We have used more robust sub-titles in the revised manuscript. The changes are compared in Response Table 4.

Response Table 4 | Previous and revised sub-titles

Previous	→	Revised
Optimal τ_{clear} , τ_{dark} , and T_{tran} for maximum building energy saving	→	Optimal key parameters for maximum building energy saving
	→	Global and regional performance evaluation

Also, we apologize for the misleading first sentence in the discussion section. In this section, we intend to discuss the limitation of our proposed models (including the tool) in the first few sentences. To avoid misunderstanding, we revised it as:

“It’s important to understand that the derived values from the ANN models are only valid under specific conditions. For instance, the optimal T_{tran} could differ when the TR layer is applied on the interior window surface rather than the exterior one.”

Reviewer #2 (Remarks to the Author):

The paper proposes a summary of an enormous investigation concerning the thermal performance of thermo-responsive dynamic windows in managing solar gains, and thus in improving thermal behavior and energy demands of buildings. The main novelty of the study is the global-scale analysis, starting from the U.S. conditions, carried out by developing artificial neural network models for suitably predict performance indicators of TR windows, also in terms of world heatmaps indicators for TC windows, and thus allowing a powerful tool for simulating the thermal performance of TR windows in many geographical sites. More than two thousand global weather conditions have been used for the training of ANNs. Also, the authors provide software to optimize TR materials, consisting of an open-source mapping tool. About the results of the study, these are very wide and interesting, following a large analysis of outcomes from more than 2.8 million simulations (considering all main technologies and materials, and thus peculiarities), and these revealed the usefulness of thermo-responsive windows mainly for buildings characterized by both significant energy demands for space heating and space cooling. It should be noted that, for reason of brevity, merely few results and good discussion are provided in the main manuscript, while much other info, data, figures and tables are reported in the supplementary materials really worthy and meaningful. My general opinion about this manuscript is surely favorable. It concerns an important topic, a useful arrow to improve, in future years, thermal and energy performance of both new and existing buildings, by proposing a method and a technology that can be applied worldwide, without limitation of architectural technology or construction techniques, but suitable (or not) only based on technical and thermodynamic considerations, for which the proposed methods (and tools) are an important support to use and thus to decide. That's why I am favorable to this study: it is global and purpose, besides some results, also a tool to apply and check such technologies in different locations and by optimizing the configuration.

We appreciate the reviewer's positive feedback on our work. We are also pleased to note that the reviewer found the proposed method impactful and the supplementary information valuable and meaningful.

I have some minor suggestions for the authors:

- At line 29, it could be worthy to cite the impacts of buildings on the energy demands and mandatory policies (if any) for improving the energy performance of the construction sector in different world areas, e.g., Europe, North and South America, Asia, Oceania and Africa. This may be useful to understand the impacts of buildings on energy demands at a global scale. Indeed, the global scale is the “spatial domain” of this investigation.

We are grateful to see that the reviewer read our introduction carefully. To include different world areas, we replace the previous Reference 2 (an IEA report) with a new reference, which is an IPCC report that includes global and regional analysis. We also added a most recent report focusing on

the U.S. building sector. To avoid lumped references, we spread the citations in the revised sentence:

“This is particularly relevant as energy demand rebounds in the post-pandemic era⁴, alongside the escalation of extreme weather events⁵ and the implementation of stringent decarbonization policies^{2,6}.”

- The introduction of the paper: some lumped references do not allow the specific contribution of each cited paper. For instance, check 8-19 (line 34): the sentence used to cite such papers is too general.

We appreciate the reviewer’s suggestion. In Line 34, we cited three articles (Reference 8-10), which are review papers or general discussions on thermochromic windows. To avoid confusion, we delete Reference 8, which is more focusing on material discussions.

“This self-regulating mechanism, requiring no manual intervention, curtails the energy expended on heating, ventilation, and air conditioning (HVAC) systems⁹⁻¹⁰.”

- The introduction describes the main properties and features of TC windows and thus the clear-state solar transmittance, the dark-state solar transmittance, the transition temperature, the available thermochromic materials and so on. There are, on the other hand, no summary information about the real use of TC windows in buildings. Is this technology spread, and used, in the sector? How much and in which countries it is common?

We appreciate the reviewer’s question about the real use of TC windows in buildings. In the revised supplementary information, we add a summary of experimental study on full-size TC windows in real building environment in Supplementary Note 1.

“In most material research, it is highly challenging to fabricate full-sized TR windows and conduct experiments in actual buildings, especially for labs that can only produce small-size samples. Instead, most studies experimented their samples in small-scale chambers (see Table 2 in *Cell Reports Physical Science* 4.5 (2023)) or evaluating the energy performance of materials in buildings through simulations. Only a few studies have employed full-sized TR windows in real buildings for field experiments (summarized in Supplementary Table 4).

A publication in 2013 introduced an experimental work on large-area polymer thermochromic (TC) laminated windows in a full-scale testbed office in our lab [*Solar Energy Materials and Solar Cells* 116 (2013): 14-26]. Several measured parameters, such as transmitted solar radiation, incident vertical irradiance, and outdoor dry-bulb temperature, were compared against the predicted values. EnergyPlus was used to calculate the annual energy performance for Chicago

and Houston. However, the thermochromic module in EnergyPlus was not validated in this study. Both empirical and simulation data were utilized to demonstrate that the ideal critical switching temperature for TC windows should be determined by the zone heat balance rather than the ambient air temperature. Another demonstration work was also reported in 2013 by our lab regarding the energy performance of electrochromic and thermochromic windows in a federal building in Denver [GSA report (2013)]. The study on thermochromic windows involved several key measurements to evaluate their performance, including glazing temperature, vertical irradiance and illuminance. Environmental factors like incident solar radiation, outdoor air temperature, wind, and indoor air temperature were also monitored to understand their influence on the window's switching behavior. EnergyPlus simulation was conducted for annual energy calculation and was not validated by measured data. In 2013 and 2015, two publications reported the demonstration and simulation of VO₂ single glazing [Solar energy materials and solar cells 117 (2013): 168-173] and double glazing [Solar Energy 120 (2015): 55-64] in a full-scale room in Hefei, China, respectively by the same research team. The simulated cooling load was validated by the measured data. However, the simulation tool is BuildingEnergy software, which is more user-friendly, but less customizable compared with EnergyPlus. They concluded that the TC windows were suitable for hot climates rather than cold climates. VO₂ double window consumes approximately 11.1% less cooling energy than that with an ordinary double window. Another paper published in 2015 reported testing thermotropic glazing and a triple glazing unit in a full-scale outdoor test cell to evaluate their thermal and optical performance [Proceedings of building simulation (2015)]. Measurements were taken over several days under varying solar radiation and temperature conditions, focusing on internal glass surface temperature, transmitted solar radiation, and heat flux. The experimental data were then compared with simulation results from EnergyPlus to assess the accuracy of the built-in thermochromic model. Results showed that the errors of simulated internal glazing surface temperature, transmitted solar radiation, and heat flux stayed in acceptable range, proving the accuracy of the built-in thermochromic model in EnergyPlus. This study will be introduced in detail later in the Credibility Analysis Section (B).

Supplementary Table 4|Summary of experimental study on full-size TC windows in real building environment

Publishing year	TC material	Building	Location	Simulation software	Validation	Reference
2013	Polymer	Full-scale, south-facing, conditioned testbed office	Berkeley, California, USA	EnergyPlus	No	Solar Energy Materials and Solar Cells 116 (2013): 14-26
2013	Not mentioned	9,500-ft ² perimeter zone of a Federal office building	Denver Federal Center, Colorado, USA	EnergyPlus	No	GSA report (2013)

2013	VO ₂	2.9 x 1.8 x 1.8 m ³	Hefei, China	BuildingEnergy	Yes	Solar energy materials and solar cells 117 (2013): 168-173
2015	VO ₂	2.9 x 1.8 x 1.8 m ³	Hefei, China	BuildingEnergy	Yes	Solar Energy 120 (2015): 55-64
2015	Technology based on a coreshell particle suspension	1.6 x 3.6 x 2.5 m ³ TWINS outdoor test cell	Torino, Italy	EnergyPlus	Yes	Proceedings of building simulation (2015)

”

In terms of commercially available products, TC windows are not widely spread at this point. Two products were available on the U.S. market that were listed in the International Glazing Database (IGDB):

1. Ravenbrick that was tested in the GSA building went bankrupt
2. Pleotint/Suntuitive shows the message in their website "Due to COVID-19's impact on our operations, Pleotint, LLC has decided to temporarily suspend all production and sales activities of its Suntuitive® product line."

Please note that the market of new building technologies grows slow, even though some technologies show great energy saving potentials. It took approximately 30 years from the introduction of Low-E windows in the 1980s for the technology to become mature and widely adopted in the market.

- At line 92, it is written that a comprehensive guideline is provided. Please, can you specify in which section of the paper is it?

The original sentence is “A comprehensive understanding of such correlations is **paramount** for optimizing dynamic window materials that suit for diverse weather conditions, or customizing materials and dynamic windows for specific locations.” The intended meaning we wanted to convey was “comprehensive understanding of such correlations is paramount”, not “comprehensive guideline is provided”. In this sentence, “such correlations” indicates “the correlations between climate suitability and intrinsic material properties on a worldwide scale”, which is the main topic discussed in the study.

To avoid misunderstanding, we change “paramount” into “important” in the revised manuscript.

“A comprehensive understanding of such correlations is important for optimizing dynamic window materials that suit for diverse weather conditions, or customizing materials and dynamic windows for specific locations.”

- Phoenix, Baltimore, and Minneapolis, according to the authors, are cities representing typical climates (hot, mixed, and cold). Please, may you provide more data? For instance, the Köppen-Geiger classification, HDD and CDD and so on?

We appreciate the reviewer's suggestions. In the revised Supplementary Table 3, we add climate info including Köppen-Geiger classification, HDD (Heating Degree Days) and CDD (Cooling Degree Days). The HDD and CDD data were obtained from the STAT files in the EnergyPlus weather data.

Supplementary Table 3 | U.S. Climate zone classification and representative cities³

Climate zone	Representative city	Köppen-Geiger classification	Heating Degree Days (HDD)*	Cooling Degree Days (CDD)*
1A	Miami, Florida	Am	72	2477
2A	Houston, Texas	Cfa	786	1667
2B	Phoenix, Arizona	BWh	523	2532
3A	Atlanta, Georgia	Cfa	1497	1023
3B-CA	Los Angeles, California	BSh and Csa/Csb	713	343
3B-other	Las Vegas, Nevada	BWh	1169	1860
3C	San Francisco, California	Csb	1504	79
4A	Baltimore, Maryland	Cfa	2537	682
4B	Albuquerque, New Mexico	BSk	2261	749
4C	Seattle, Washington	Csb	2627	98
5A	Chicago, Illinois	Dfa	3506	468
5B	Denver, Colorado	BSk	3301	432
6A	Minneapolis, Minnesota	Dfa	4203	417
6B	Helena, Montana	BSk	4266	208
7	Duluth, Minnesota	Dfb	5236	116
8	Fairbanks, Alaska	Dfc	7516	39

* Annual (standard) heating/cooling degree-days (18.3°C (or 65°F) baseline)

- Many symbols, parameters, and acronyms are used. Many times, I have searched for the meaning of a symbol, without an immediate understanding. Please, provide a full nomenclature, with all parameters and SI units.

We appreciate the reviewer's suggestion. A nomenclature is now added the supplementary information.

Nomenclature

Abbreviation [unit]	Explanation
τ_{clear} [-]	Clear-state solar transmittance
τ_{dark} [-]	Dark-state solar transmittance
T_{tran} [°C]	Transition temperature
$\Delta\tau_{sol}$ [-]	Solar modulation, i.e., $\tau_{dark} - \tau_{clear}$
ΔE_{total} [kWh m ⁻²]	Total site energy saving per conditioned building (floor) area
E_{total} [kWh m ⁻²]	Total site energy per conditioned building (floor) area
ΔE [kWh m ⁻²]	Energy saving per conditioned building (floor) area
ΔE_{TR} [kWh m ⁻²]	Total site energy saving per conditioned building (floor) area of buildings using TR windows
E_{TR} [kWh m ⁻²]	Total site energy per conditioned building (floor) area of buildings using TR windows
E_{dark} [kWh m ⁻²]	E_{total} by constantly dark (low T_{tran}) TR windows
E_{clear} [kWh m ⁻²]	E_{total} by constantly clear (high T_{tran}) TR windows
ΔE_{dark} [kWh m ⁻²]	ΔE_{total} by constantly dark (low T_{tran}) TR windows
ΔE_{clear} [kWh m ⁻²]	ΔE_{total} by constantly clear (high T_{tran}) TR windows
E_{static} [kWh m ⁻²]	The smaller between E_{dark} and E_{clear} , i.e., $\min(E_{dark}, E_{clear})$
ΔE_n [kWh m ⁻²]	The necessity level of using TR windows compared to static windows, defined as $E_{static} - E_{TR}$
ReLU	Rectified Linear Unit, a piecewise linear function that outputs the input directly if it is positive, otherwise, outputs zero.
TRRI [kWh ² m ⁻⁴]	$ReLU(\Delta E_{TR}) \times ReLU(\Delta E_n)$
DNI [W m ⁻²]	Direct normal irradiation. When it comes daily total DNI, the unit is [kWh m ⁻²]
GHI [W m ⁻²]	Global horizontal irradiation. When it comes daily total GHI, the unit is [kWh m ⁻²]
DHI [W m ⁻²]	Diffuse horizontal irradiation. When it comes daily total DHI, the unit is [kWh m ⁻²]
GVI [W m ⁻²]	Global vertical irradiance. When it comes daily total GVI, the unit is [kWh m ⁻²]
Lat [°]	Latitude
T_{avg} [°C]	Average air temperature
T_{min} [°C]	Minimum air temperature
T_{max} [°C]	Maximum air temperature
TR	Thermo-responsive

CO ₂	Carbon dioxide
HVAC	Heating, ventilation, and air conditioning
TC	Thermochromic
VO ₂	Vanadium dioxide
NIR	Near-infrared
PNIPAM	Poly(N-isopropylacrylamide)
EC	Electrochromic
WO ₃	Tungsten oxide
MoO ₃	Molybdenum oxide
V ₂ O ₅	Vanadium oxide
RME	Reversible metal electrodeposition
SPD	Suspended particle device
PDLC	Polymer dispersed liquid crystal
LC	Lighting control
DOE	Department of Energy
ANN	Artificial neural network
low-e	Low-emissivity
SHGC	Solar heat gain coefficient
DX	Direct expansion
MLP	Multi-Layer Perceptron
BTO	Building Technologies Office
BENEFIT	Buildings Energy Efficiency Frontiers & Innovation Technologies

- Figure 3 may be more readable if readers can immediately understand the climate of such locations. Could you provide a map with HDD and CDD or similar indicators of the coldness and warmness of the U.S. locations?

- At the world level (figure 4), more or less the climates are clear, so the previous request is not so mandatory.

We appreciate the reviewer’s concern about this issue. The representative cities in this study are chosen according to the U.S. climate classification for building energy codes and standards (Reference 35 in the manuscript). We now added a map to the Supplementary Table 3 to help readers better understand the results.

“**See the map below to understand the climate zone classification in the U.S.³. The numbers 1 – 8 indicate very hot, hot, warm, mixed, cool, cold, very cold, and subarctic, respectively.

”

- I suggest rephrasing at line 353 (Visual comfort is not within the scope of the study and can be found elsewhere). For sure it can be found elsewhere but probably the results will be different. So, please rephrase, now it seems that paper 40 is connected to this one.

We appreciate the reviewer pointing this out. We revised this sentence as below:

“Visual comfort is beyond the scope of this study, and while related information can be found in other sources⁴⁰, the results may vary depending on the specific context and methodology used.”

- Just a curiosity concerning Figure 4, last picture (down and right, 4h). How do the authors explain that Mediterranean countries have so low optimal transition temperature (around 5 °C) and this is valid mainly for the African coasts of the Mediterranean and not for the European coast (where higher optimal T_{tran} occur)?

We are glad that the reviewer noticed the difference of optimal transition temperature (T_{tran}) among the Mediterranean countries.

First, a very low T_{tran} in a TR window indicates that the window barely turns clear under solar radiation in warm days. In other words, the low T_{tran} enables the TR window stay dark and block the solar radiation.

Second, the optimal T_{tran} of TR window is determined by latitude, GVI, air temperatures and solar transmittances according to the ANN structure (Supplementary Figure 118c). For fixed solar

transmittances of certain TC material ($\tau_{dark} = 0$, $\tau_{clear} = 0.8$ in the case of Fig. 4h), the optimal T_{tran} is mainly influenced by the climate factors, including latitude, GVI, and air temperatures.

Third, let's compare those climate factors between the African and European coasts of the Mediterranean.

(A) Latitude

Apparently, European coast has higher latitude than the African side, leading to certain climate variances.

(B) GVI (global vertical irradiance)

As shown in Supplementary Figure 78a (also attached below), the GVI in the African coast is higher than that in the European side. This could be a main reason for the distinct optimal T_{tran} . A low T_{tran} keeps the TR window in dark state, therefore reducing the cooling energy in the African coast. We also noticed that the regions with low T_{tran} usually have high GVI, globally.

(C) Air temperatures

As shown in Supplementary Figure 77 (also attached below), the African coast has higher average and maximum air temperatures compared to the European side. Generally, the buildings on the African side of the Mediterranean require more cooling energy throughout the year. The hot, arid conditions, especially in regions near the Sahara, lead to a significant demand for air conditioning to maintain comfortable indoor environments. The mild winters in the African Mediterranean regions result in relatively low heating energy requirements. That's why the optimal T_{tran} is low in these regions, keeping the TR window dark to reduce solar heat gain. On the other hand, the European side experiences more temperate and cooler climates, especially in the northern regions, leading to a lower demand for cooling energy and a higher demand for heating energy. It requires the TR window switching at optimal T_{tran} to balance the heating and cooling energy saving. Also, the necessity level of using TR windows is higher in the European coast compared to the African side. It tends to have a zero E_n in the African coast due to no heating demand (Supplementary Figure 78c).

In summary, the distinct optimal T_{tran} of TR windows in the Mediterranean countries mainly result from various climate conditions, especially different solar radiation and air temperature, leading to different demands on heating and cooling in buildings. Due to the word limit of this article, we will not further discuss this particular topic in the manuscript.

Supplementary Figure 1|World heatmaps of global temperature distribution for ANN inputs⁵.

Supplementary Figure 2|World heatmaps of GVI, optimal ΔE_{TR} , ΔE_n , and TRRI generated by artificial neural networks (ANNs).

- Supplemental materials and explanations at the end of the paper are useful and meaningful, well-done.

We appreciate that the reviewer found supplementary information useful and meaningful. Many thanks for reading the long document carefully.

- Please, given that the whole study is based on simulation, provide a validation of the main energy results and outcomes of the initial 3 cities (Phoenix, Baltimore, and Minneapolis).

We appreciate that the reviewer noticed the lack of validation in our study, which was also mentioned by the other review. We have now added a paragraph of stepwise model validation in the Method, and a credibility analysis in the supplementary information.

“Methods: Stepwise model validation

Though it is challenging to directly validate our proposed models, it is feasible to validate the model step by step. We analyzed the accuracy of the EnergyPlus software, the built-in thermochromic model, ANNs and weather inputs, respectively. The credibility has been demonstrated by results from both previous literatures and our study. Detailed Credibility Analysis can be found in Supplementary Note 1.”

“Supplementary Note 1

In most material research, it is highly challenging to fabricate full-sized TR windows and conduct experiments in actual buildings, especially for labs that can only produce small-size samples. Instead, most studies experimented their samples in small-scale chambers (see Table 2 in *Cell Reports Physical Science* 4.5 (2023)) or evaluating the energy performance of materials in buildings through simulations. Only a few studies have employed full-sized TR windows in real buildings for field experiments (summarized in Supplementary Table 4).

A publication in 2013 introduced an experimental work on large-area polymer thermochromic (TC) laminated windows in a full-scale testbed office in our lab [*Solar Energy Materials and Solar Cells* 116 (2013): 14-26]. Several measured parameters, such as transmitted solar radiation, incident vertical irradiance, and outdoor dry-bulb temperature, were compared against the predicted values. EnergyPlus was used to calculate the annual energy performance for Chicago and Houston. However, the thermochromic module in EnergyPlus was not validated in this study. Both empirical and simulation data were utilized to demonstrate that the ideal critical switching temperature for TC windows should be determined by the zone heat balance rather than the ambient air temperature. Another demonstration work was also reported in 2013 by our lab regarding the energy performance of electrochromic and thermochromic windows in a federal building in Denver [*GSA report* (2013)]. The study on thermochromic windows involved several

key measurements to evaluate their performance, including glazing temperature, vertical irradiance and illuminance. Environmental factors like incident solar radiation, outdoor air temperature, wind, and indoor air temperature were also monitored to understand their influence on the window's switching behavior. EnergyPlus simulation was conducted for annual energy calculation and was not validated by measured data. In 2013 and 2015, two publications reported the demonstration and simulation of VO₂ single glazing [*Solar energy materials and solar cells* 117 (2013): 168-173] and double glazing [*Solar Energy* 120 (2015): 55-64] in a full-scale room in Hefei, China, respectively by the same research team. The simulated cooling load was validated by the measured data. However, the simulation tool is BuildingEnergy software, which is more user-friendly, but less customizable compared with EnergyPlus. They concluded that the TC windows were suitable for hot climates rather than cold climates. VO₂ double window consumes approximately 11.1% less cooling energy than that with an ordinary double window. Another paper published in 2015 reported testing thermotropic glazing and a triple glazing unit in a full-scale outdoor test cell to evaluate their thermal and optical performance [*Proceedings of building simulation (2015)*]. Measurements were taken over several days under varying solar radiation and temperature conditions, focusing on internal glass surface temperature, transmitted solar radiation, and heat flux. The experimental data were then compared with simulation results from EnergyPlus to assess the accuracy of the built-in thermochromic model. Results showed that the errors of simulated internal glazing surface temperature, transmitted solar radiation, and heat flux stayed in acceptable range, proving the accuracy of the built-in thermochromic model in EnergyPlus. This study will be introduced in detail later in the Credibility Analysis Section (B).

Supplementary Table 4|Summary of experimental study on full-size TC windows in real building environment

Publishing year	TC material	Building	Location	Simulation software	Validation	Reference
2013	Polymer	Full-scale, south-facing, conditioned testbed office	Berkeley, California, USA	EnergyPlus	No	Solar Energy Materials and Solar Cells 116 (2013): 14-26
2013	Not mentioned	9,500-ft ² perimeter zone of a Federal office building	Denver Federal Center, Colorado, USA	EnergyPlus	No	GSA report (2013)
2013	VO ₂	2.9 x 1.8 x 1.8 m ³	Hefei, China	BuildingEnergy	Yes	Solar energy materials and solar cells 117 (2013): 168-173
2015	VO ₂	2.9 x 1.8 x 1.8 m ³	Hefei, China	BuildingEnergy	Yes	Solar Energy 120 (2015): 55-64

2015	Technology based on a coreshell particle suspension	1.6 x 3.6 x 2.5 m ³ TWINS outdoor test cell	Torino, Italy	EnergyPlus	Yes	Proceedings of building simulation (2015)
------	---	--	---------------	------------	-----	--

It will be extremely challenging to directly validate the results of this study. Due to the extensive range of TR window parameters and global climate conditions covered in this research, it is only feasible to validate one or a few of these variables with real-world experiments. Moreover, the simulations in this study utilize reference buildings from the U.S. Department of Energy’s (DOE) prototype building models, chosen for their representativeness. The window data in this study were obtained from measurements of small-size samples. In practice, constructing buildings identical to the standard model and manufacturing windows identical to the sample on a large scale are both time-consuming and costly. Although direct validation is difficult, the reliability of this work can still be indirectly verified by analyzing the credibility of each component of this work.

Credibility Analysis

(A) EnergyPlus

This research primarily utilized EnergyPlus, a whole building energy simulation program. Its development is funded by the U.S. DOE Building Technologies Office (BTO). EnergyPlus has undergone extensive validation and verification processes [Testing and Validation, EnergyPlus]. It has been tested against empirical data from real buildings and benchmarked against other established simulation tools [IEA SHC Task 34/Annex 43]. It has been widely adopted in the industry and academia, with numerous peer-reviewed studies attesting to its accuracy and robustness [research articles related to EnergyPlus].

A previous study validated EnergyPlus by comparing simulation results against measured data from real-world buildings and controlled test environments [Building Simulation 2019, Vol. 16, IBPSA, 2019]. It is an empirical validation project conducted by multiple national laboratories, including Oak Ridge National Laboratory (ORNL). The validation utilized ORNL’s Flexible Research Platform, a small office building with detailed monitoring systems. The project generated extensive empirical data sets, including cooling energy consumption, fan energy consumption, and zone temperatures, under various test conditions. The accuracy of EnergyPlus was quantified using metrics such as Normalized Mean Bias Error (NMBE) and Coefficient of Variation of the Root Mean Square Error (CV(RMSE)),

$$NMBE = \frac{1}{\bar{M}} \frac{\sum_{i=1}^n (M_i - S_i)}{n} \times 100\%$$

$$CV(RMSE) = \frac{1}{\overline{M}} \sqrt{\frac{\sum_{i=1}^n (M_i - S_i)^2}{n}} \times 100\%$$

where M , S , and n represent the measurement, simulation, and the number of data, respectively. Upper bar refers to the average. The hourly NMBE and CV(RMSE) were less than 2.6% and 5.9%, respectively, indicating that simulation and experimental energy consumption are well matched.

(B) Built-in thermochromic module

Furthermore, the thermochromic module has also been verified by comparing with experimental data in a previous study [*Proceedings of building simulation (2015)*]. In this study, simulation results using EnergyPlus and the built-in thermochromic module were compared with the experimental results measured in a full-scale outdoor test facility (1.6 m x 3.6 m x 2.5 m) in Torino, Italy. The experimental data was collected using the side-by-side test cell facility equipped with a triple glazing unit (TGU) and a triple glazing unit with thermotropic (TT) glazing on the external side (TT+TGU). The data for comparison was taken from April 12th to April 15th, 2013, which included medium to high vertical solar radiation and temperature variations. Supplementary Figure 126 shows the results of the reference TGU, with good agreement between simulation and experimental data in general. Simulated and measured parameters include internal surface temperature of the glazing ($T_{glass,in}$), transmitted solar radiation (G_{in}), and heat flux (radiative longwave and convective) on the internal surface of the glazing (HF_{lw}). A two-hour delay was observed in measured temperatures compared to simulation due to EnergyPlus not accounting for the thermal mass of the glazing. This issue could be solved by using equivalent models as demonstrated in our previous work [*Applied Energy 301 (2021): 117467*]. Please note that here our parametric study on TC windows doesn't take the thermal mass of the glazing into account. A peak difference in the heat flux in the afternoon was also observed, likely due to the heat flow meter sensor overheating from direct solar radiation [*Proceedings of building simulation (2015)*].

Supplementary Figure 126|Simulation and experimental results of the reference TGU. a. Transmitted solar radiation (left y axis), and internal surface temperature of the glazing (right y axis). b. Heat flux on the internal surface of the glazing. Figure adapted from [*Proceedings of building simulation (2015)*].

To quantitatively evaluate the simulation performance, the errors between simulation and experimental data were defined as below:

Mean Bias Error (MBE):

$$MBE = \frac{1}{n} \sum_{i=1}^n (X_{mod} - X_{exp})$$

Root Mean Square Error (RMSE):

$$RMSE = \sqrt{\frac{1}{n} \sum_{i=1}^n (X_{mod} - X_{exp})^2}$$

Percentage Root Mean Square Error (PRMSE):

$$PRMSE = \sqrt{\frac{1}{n} \sum_{i=1}^n \left(\frac{X_{mod} - X_{exp}}{X_{exp}} \right)^2}$$

where n is the number of measurements.

The results are listed in Supplementary Table 5. The errors remain within an acceptable range, demonstrating that the simulation results for the reference room using EnergyPlus are reliable.

Supplementary Table 5|Simulation errors compared with measured results in the cell with TGU [*Proceedings of building simulation (2015)*]

	MBE	RMSE	PRMSE
$T_{glass,in}$ (°C)	-0.5	1.5	5.1%
G_{in} (W m ⁻²)	-0.6	11.5	-
HF_{lw} (W m ⁻²)	-6.3	13.7	-

In Supplementary Figure 127, “TT+TGU_E+” indicates the simulation results of triple glazing unit with thermotropic glazing using the built-in thermochromic model. “EMS – E+” indicates an alternative EMS (Energy Management System) EnergyPlus model, which will not be considered in this discussion. The TT+TGU simulation showed good alignment with measured solar radiation data but exhibited a 2-hour delay and peak temperature differences of 3-4°C during peak solar radiation hours. Additionally, heat flux discrepancies of 10-15 W/m² were noted. These discrepancies, primarily due to increased solar reflectance of the TT glazing in experimental conditions, needed a model calibration to better match the experimental data. The calibration

involved increasing the solar and luminous reflectance by a constant factor, which significantly improved model accuracy (denoted with “_mod” in Supplementary Figure 127). After calibration, the MBE for $T_{glass,in}$ was reduced to -0.01°C , with a RMSE of 1.38°C and a PRMSE of 5%. Similarly, for the HF_{hw} , the MBE was reduced to -0.84 W/m^2 , and the RMSE to 10.31 W/m^2 . These values indicate that the calibrated built-in TC model in EnergyPlus is reliable for predicting the performance of TC glazing technologies.

Supplementary Figure 127|Simulation and experimental results of TT+TGU. a. Internal surface temperature of the glazing. b. Heat flux on the internal surface of the glazing. c. Transmitted solar radiation. d. Table of simulation performance of different methods. Figure adapted from [Proceedings of building simulation (2015)].

(C) ANNs

Our study reported multiple different artificial neural networks (ANNs), each trained and tested using different inputs and outputs. Here we select several representative ANNs and their performance, which are displayed in Supplementary Figure 128. The training of each neural network was based on the mean squared error (MSE), defined as

$$MSE = \frac{1}{n} \sum_{i=1}^n (S_i - M_i)^2,$$

where S and M represent the predicted and actual values, respectively. Based on MSE values, we calculated other error metrics, which are listed in Supplementary Table 6. Different ANN structures

exhibit various performances as shown in Supplementary Figure 128. Generally, the more effective data can be used to train the ANN, the better its performance becomes. It is noteworthy that due to the inherent randomness in neural network training, the same ANN structure may yield different MSE values across different trainings. In this study, we selected the best-performing results from multiple trainings. Although errors are inevitable, the linear regression between the actual and predicted values of the ANNs is obvious, enabling the ANNs to provide valuable guidance for the design of the TR window. From the error metrics, we can conclude that the errors of the ANNs in this study stay within an acceptable range (Supplementary Table 6).

Supplementary Figure 74a

Figure 4a

Supplementary Figure 122a

Supplementary Figure 123b

Supplementary Figure 122b

Supplementary Figure 124b

Supplementary Figure 118c

Supplementary Figure 121b

Supplementary Figure 128|Four selected ANNs and the performance of training and testing. Figures are adapted from manuscript and SI.

Supplementary Table 6|Testing performance of four selected ANNs in this study

Output of ANN	MSE	RMSE	CV(RMSE)
GVI	0.0080	0.0894	3.4%
ΔE_{TR} (trained by all conditions)	0.0160	0.1265	1.3%
ΔE_n (trained by all conditions)	0.0144	0.1200	3.8%
Optimal T_{tran} (trained by the optimal conditions)	2.4458	1.5639	7.9%

(D) Input data of solar radiation and air temperature

Solar radiation

Solar radiation inputs of this study include direct normal irradiation (DNI), global horizontal irradiation (GHI), and diffuse horizontal irradiation (DHI), which are obtained from the Solargis solar radiation model [Global Solar Atlas]. In a validation report, the accuracy of solar radiation data was calculated through the comparison with ground-data from the reference stations [Global Solar Atlas 2.0 : Validation Report]. In this report, the solar radiation data has been validated at 228 public sites worldwide. The bias (in percent) of GHI and DNI are shown in Supplementary Figure 129 and 130, respectively. The mean bias of GHI and DNI for all sites are 0.3% and 2.2%, respectively. Besides the validation in this report, a list of independent validation studies can also be found in Section 6 of this report. The Solargis solar radiation model has been proved reliable and the solar radiation data used in this study falls within an acceptable error range.

Supplementary Figure 129|GHI bias on the background of climate zones (values in percent). Climatic classes: A – tropical; B – arid; C – temperate; D – cold; E – polar. Figure adapted from [Global Solar Atlas 2.0 : Validation Report].

Supplementary Figure 130|DNI bias on the background of climate zones (values in percent). Figure adapted from [Global Solar Atlas 2.0 : Validation Report].

Air temperature

Global air temperature input of this study include minimum, mean, and maximum temperature, which are obtained from WorldClim 2 database [*International journal of climatology* **37**, 4302–4315 (2017)]. The accuracy of the dataset was demonstrated by comparing with station data. RMSE values of minimum, maximum, and average temperature are shown in Supplementary Figure 131a-c. All temperature variables exhibited a global correlation coefficient of 0.99 or higher between estimated and observed values, and an average RMSE ranging from 1.1 to 1.4°C (Supplementary Figure 131d). The accuracy is deemed sufficient to predict the trend of building energy performance in a global scale.

Supplementary Figure 131|a-c. Spatially aggregated RMSE values of minimum (a), maximum (b), and average (c) temperature. d. Table of global cross-validation statistics for temperature models. Figures adapted from [*International journal of climatology* 37, 4302–4315 (2017)].

In summary, the accuracy of the EnergyPlus software, the built-in thermochromic model, ANNs and weather inputs are analyzed here. The performance of each component in our proposed models are summarized in Supplementary Figure 132. As George E. P. Box. said, "All models are wrong, but some are useful." We demonstrated that our models effectively predicted the trend of building energy saving and necessity level of using TR windows.

Supplementary Figure 132|Summary of accuracy performance of each component in the proposed models of thermo-responsive dynamic windows.

- Add limitations and future perspectives. For instance, the next study may be based on validation and comparison against experimental results.

We appreciate the reviewer's suggestions. In the previous response, we answered why the direct validation is challenging, and we provide a stepwise model validation instead. Therefore, our next study will not focus on validation and experiments. We did mention next steps in our manuscript: "ANN models with more variables (e.g., building types, window orientations, etc.) and more indicators (e.g., total source energy) will be trained in the future and new versions will be released upon request." This could also be interpreted as the limitations of this study. We also mentioned other limitations in the Discussion section, such as "the optimal T_{tran} could differ when the TR layer is applied on the interior window surface rather than the exterior one", and "... this study mainly considered building energy savings of TR windows. Visual comfort is beyond the scope of this study, ...". Considering the word limit for this article, we will not further extend limitations and future perspectives in the manuscript.

Finally, in my opinion, this is a comprehensive paper, novel, providing interesting results and giving contribution to the research community involved in the field. I recommend a minor revision.

Again, we appreciate the reviewer's positive feedback and valuable suggestions.

Point-by-point actions in response to the Editors' and Reviewers' comments

Manuscript # NCOMMS-24-18413A

We sincerely thank the editors and reviewers for their valuable comments in this review process. Following are the detailed actions taken in light of editors' and reviewers' comments:

Reviewer #1 (Remarks to the Author):

I would like to thank the authors for their thorough review of the manuscript and the detailed response to my comments. I believe that the manuscript has improved significantly since it was first submitted.

I insist that this is a significant piece of work that should be published, due to its usefulness to the industry and academia.

We sincerely appreciate your positive feedback on our work. It is gratifying to know that our efforts have been well received.

However, I will also need to insist that any paper that is based on simulations needs to validate its results. I realise the difficulties of reproducing the buildings, or having full-sized samples. I also appreciate the fact that the tool used (EnergyPlus), as well as the rest of the components are all individually validated and/or commonly used. The validity of each of the components, however, does not guarantee the validity of the final product. All models are wrong, but the non-validated ones can be more wrong and potentially not useful.

It is my belief that the authors should either find a way to experimentally validate their results or propose another methodology to prove that the tool provides accurate results.

We highly respect the reviewer's insistence on scientific rigor and the dedication to ensuring the validity of the findings. We also agree with the reviewer that direct validation is crucial to ensuring the accuracy of our models. As explained in SI and the previous response letter, the difficulties of direct validation are three-fold:

1. It is only feasible to validate one or a few of locations with real-world experiments, instead of the global results in this study. The real-world climate conditions could be different from those in the climate database, which is an average of long-term measurement.
2. The simulations in this study utilize reference buildings from the U.S. Department of Energy's (DOE) prototype building models, chosen for their representativeness. At this moment, we don't have the same or similar buildings for field experiments.

3. The window data used in this study were derived from measurements of small-sized samples. In practice, producing TR windows identical to the sample on a large scale would be both time-intensive and costly.

At this stage, direct validation of our models is difficult to provide in a short time. However, this does not diminish the importance or necessity of direct validation. We hope to conduct field experiments under specific climate conditions when the experimental conditions allow in the future. In the revise SI, we remove George's discussion about the model accuracy, and revise the last paragraph of Credibility Analysis as

“In summary, the accuracy of the EnergyPlus software, the built-in thermochromic model, ANNs and weather inputs are analyzed here. The performance of each component in our proposed models are summarized in Supplementary Figure 127. We have successfully demonstrated the accuracy of each component of our models. Strictly speaking, the accuracy of each individual model does not fully validate the accuracy of the overall model. However, we believe this is currently the most convincing evidence we can provide. We have explained previously why it is very difficult to validate the whole model by conducting field experiments. In the future, if conditions allow, we hope to conduct field experiments under specific climate conditions to further test the whole model's accuracy.”

Reviewer #2 (Remarks to the Author):

Review of NCOMMS-24-18413-T, entitled “Global and regional perspectives on optimizing thermo-responsive 2 dynamic windows for energy-efficient buildings” submitted after revision to be considered for publication in Springer Nature Communications.

To the Authors

An extensive investigation is proposed to review the thermal performance of thermo-responsive dynamic windows, the management of solar gains, and thus, the improvement of thermal behavior and energy demands of buildings. A novel global-scale analysis is proposed, even by developing artificial neural network models for suitably predict performance indicators of TR windows; thus, a powerful tool for simulating the thermal performance of TR windows in many geographical sites is developed, and the use of the framework, by applying millions of simulations, provided many interesting results and a worthy discussion. I have had a very positive opinion already concerning the original submission. The method is sound, the investigation is novel, conclusions are supported by data. Really, I asked for some deeper info and comments, some better explanations (for instance, about climates, HDD, and CDD), some additional discussion, avoiding of lumped references, comments about validations, limits of the present investigation (by commenting about these, if any). The authors replied carefully, adding both new parts in the manuscript and the supplementary

materials. They answered my comments and curiosities, and everything was clear, careful, and meaningful. My opinion is that the current version is wide, large, interesting, and worthy to be published in this prestigious journal. Accept.

Thank you for your positive evaluation of our work. Your expertise and detailed comments have greatly enhanced the quality of this manuscript.